# Microstructural constraints on magmatic mushes under Kīlauea Volcano, Hawai'i

Penny E. Wieser [1]*, Marie Edmonds [1], John Maclennan[1] & John Wheeler [2]

Distorted olivines of enigmatic origin are ubiquitous in erupted products from a wide range of volcanic systems (e.g., Hawai'i, Iceland, Andes). Investigation of these features at Kīlauea Volcano, Hawai'i, using an integrative crystallographic and chemical approach places quantitative constraints on mush pile thicknesses. Electron backscatter diffraction (EBSD) reveals that the microstructural features of distorted olivines, whose chemical composition is distinct from undistorted olivines, are remarkably similar to olivines within deformed mantle peridotites, but inconsistent with an origin from dendritic growth. This, alongside the spatial distribution of distorted grains and the absence of adcumulate textures, suggests that olivines were deformed within melt-rich mush piles accumulating within the summit reservoir. Quantitative analysis of subgrain geometry reveals that olivines experienced differential stresses of ~3–12 MPa, consistent with their storage in mush piles with thicknesses of a few hundred metres. Overall, our microstructural analysis of erupted crystals provides novel insights into mush-rich magmatic systems.

[1] Department of Earth Sciences, University of Cambridge, Downing Street, Cambridge CB2 3EQ, UK. [2] Department of Earth, Ocean and Ecological Sciences, University of Liverpool, Liverpool, UK. *email: pew26@cam.ac.uk

Studies of the textural and geochemical characteristics of olivine crystals have revealed significant complexity in basaltic volcanic plumbing systems, including the timescales of eruptive triggering from diffusion profiles[1], storage times from crystal chemistry[2], and depths of magma storage from melt inclusion geochemistry[3]. Olivines with prominent lattice distortions are commonplace in olivine-rich basalts from a range of tectonic settings[2,4,5]. As the magmatic processes producing distorted olivines are poorly understood, these features represent an underexploited source of information.

At Kīlauea Volcano, distorted olivines are commonly attributed to ductile creep within dunitic bodies located around the central conduit[6] or within the deep rift zones (~5–9 km depth)[7,8]. However, a recent suggestion that lattice distortions are produced by an early phase of branching dendritic growth, followed by textural ripening and the merging of misoriented crystal buds, has gained considerable traction[5,9]. Lattice distortions have also been attributed to collisions between crystals during magma flow through constricted or irregular conduits[10,11].

Establishing the magmatic processes producing distorted olivines is vital to further our understanding of basaltic plumbing systems. Extracting the information held within these features is particularly crucial at Kīlauea, where olivine is the only silicate phase in magmas containing >6.8 wt% $MgO$[12] so provides the main record of pre-eruptive storage. Furthermore, Kīlauean primary melts have ~15–17 wt% $MgO$[13], yet erupted lavas contain 5–10 wt% MgO, and few crystals (1–3 vol.%)[14]. This disparity in MgO contents requires fractionation and storage of considerable volumes of olivine within the volcanic edifice[15].

Materials with high seismic velocities (interpreted as dunitic bodies) have been imaged within the summit region and deep rift zones of Kīlauea Volcano[16]. Large earthquakes, slow slip events, and landslides have been attributed to the presence of these weak dunitic bodies separating the south flank of Kīlauea from the rest of the island of Hawai'i[15,17]. Yet, the relationship between distorted olivines in erupted products and seismically-imaged dunites remains unresolved. If lattice distortions are created within dunitic bodies, microstructural interrogation may provide insights into the instability of the south flank. However, if distortions are in fact growth features, the association between dunitic bodies and edifice instability at ocean islands (e.g., Hawai'i, Reunion) may have been overestimated from petrological observations of distorted grains[5]. The hypothesis that distorted olivines originate from dunitic bodies has fueled suggestions that Kīlauea's deep rift zones allow magma to bypass the summit reservoir[8]. Re-assessing the petrological evidence for summit bypass is vital for accurate estimates of magma supply rates and $CO_2$ fluxes at Kīlauea[18].

If distorted olivines record plastic deformation, rather than crystal growth processes, their microstructures can be evaluated within the conceptual framework developed through the extensive study of naturally- and experimentally-deformed mantle peridotites. The movement of dislocations is one of the main mechanisms accommodating plastic deformation within crystals. Dislocations can be characterized in terms of a Burgers vector ($\vec{b}$) and a line direction ($\vec{u}$). The plane containing these vectors defines the slip plane[19]. The angle between $\vec{b}$ and $\vec{u}$ determines whether the dislocation is an edge dislocation ($\vec{b} \perp \vec{u}$), a screw dislocation ($\vec{b} \parallel \vec{u}$), or a mixed dislocation ($\vec{b}$ and $\vec{u}$ neither parallel or perpendicular). A slip system is described in terms of a set of slip planes, and a Burgers vector direction on which dislocation motion occurs. For example, the olivine slip system (010)[100] denotes the movement of edge dislocations along the (010) plane with a [100] Burgers vector direction. Previous work

has identified five olivine fabric types (A to E-types) from different crystallographically preferred orientations (CPOs), and determined the role that pressure, temperature, differential stress and hydration exert on their formation[20–22]. However, unlike thin sections of mantle peridotites, the orientations of erupted olivines in tephra and spatter deposits have been randomised upon eruption and again during sample preparation, so CPOs cannot be used to infer deformation conditions in volcanic olivines. Fortunately, CPOs are macroscopic features resulting from the activities of different dislocation types, which can be classified in terms of slip systems[21,23–26].

Until now, most assessments of lattice distortion in volcanic olivines have used optical microscopy[6,8,27]. This method is largely qualitative, and optical evidence for distortion is often ambiguous in small grains, or across subgrain boundaries with small misorientations (see Supplementary Figs. 1–5)[28]. A few studies have employed more quantitative methods at Kīlauea, but these have been restricted to visual examination of dislocation structures and densities following oxidation decoration of dislocations[29,30], and examination of lattice strain using in situ X-ray diffraction[28]. In this study, subgrain boundaries in individual olivine crystals within basaltic spatter and scoria from Kīlauean eruptions were mapped using electron backscatter diffraction (EBSD). This method not only offers significant improvements in spatial and angular resolution compared to optical observations (Supplementary Figs. 1–5)[31], it also provides sufficient crystallographic information to quantify the Burgers vector and slip plane, and thus the dominant slip system (see Methods section). As many deformation experiments on mantle-derived olivines were conducted at pressures and temperatures similar to those found within igneous plumbing systems (e.g., 250–300 MPa, 1200 °C)[32], assessment of the activities of different slip systems in erupted olivine crystals places constraints on the conditions of deformation (using established links between slip systems, CPOs and fabric types, and between fabric types and deformation conditions)[21,23–26]. Finally, to evaluate the suggestion that lattice distortion results from the textural maturation of olivine dendrites, we assess the crystallographic signatures of branching dendritic growth in olivine dendrites from West Greenland picrites[33,34], and compare these to microstructures in distorted Kīlauean olivines.

We demonstrate that distorted olivines show microstructures which are remarkably similar to those observed in naturally-deformed and experimentally-deformed mantle peridotites, but distinct from the crystallographic signatures of dendritic growth. These findings, alongside the spatial distribution of distorted olivines, indicate that lattice distortions formed during high temperature creep in melt-rich olivine mush piles at the base of the summit reservoir.

## Results and discussion
**Incidence of lattice distortions**. We examine olivines from a wide range of eruptions temporally associated with activity at Mauna Ulu (May 1969–December 1974). This five year period is distinct in Kīlauea's history; eruptions occurred almost simultaneously on the East and South West rift zones (ERZ; SWRZ), and at discrete sites within and around the summit caldera (Fig. 1)[35]. Thus, this suite of eruptions provides a unique opportunity to decipher the spatial distribution of distorted olivines within a single eruptive period at Kīlauea (see Supplementary Table 1 for sample details). We also investigate two further SWRZ eruptions; the circumferential lava flow on the seismic SWRZ (~1700CE; hereafter CE omitted for brevity)[36] and the Mauna Iki eruption on the volcanic SWRZ (1919–1920)[37].

Up to now, distorted olivines were thought to be reasonably rare in subaerial eruptions compared to submarine eruptions at

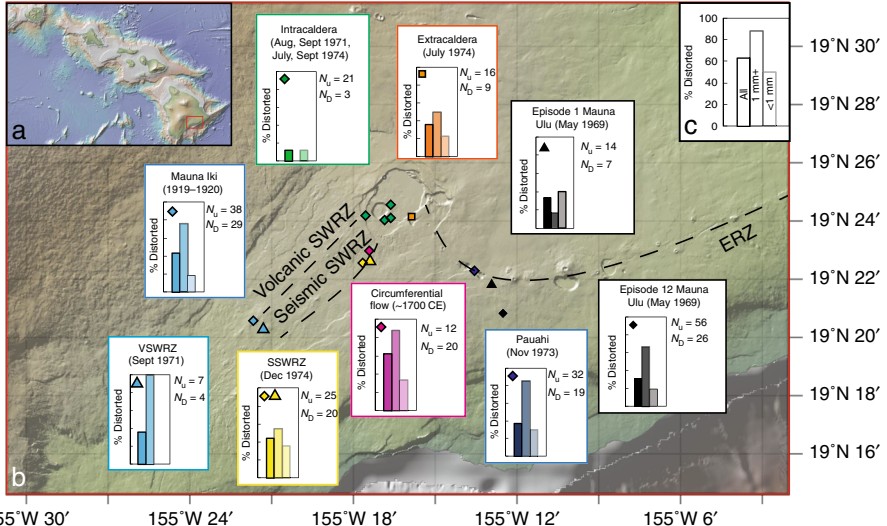

**Fig. 1 Location of eruptions examined in this study. a** Map of the Hawaiian Island Chain, with the red rectangle showing the location of Kīlauea Volcano on the island of Hawai'i (expanded in part **b**). **b** Expanded map of Kīlauea Volcano, with colored symbols showing the location of eruption deposits examined in this study (symbol colors and shapes are the same as those used in Fig. 5). The East Rift Zone (ERZ), and volcanic and seismic strand of the South West Rift Zone (VSWRZ and SSWRZ, respectively) are shown in black dashed lines. **c** Schematic histogram representing the proportion of distorted grains in three categories (all grains, grains >1 mm and <1 mm based on sieved size fractions). For each eruption site, a colored histogram following this schematic shows the proportion of distorted grains in each size category. The number of distorted ($N_D$) and undistorted ($N_U$) grains is also shown, along with the symbol used on the basemap. Basemap from GEOMAPP APP (see ref. [83]; http://www.geomapapp.org), showing the topography of Kīlauea before the onset of the 2018 summit collapse.

Kīlauea[6,15,38]. Yet, we find abundant distorted grains in subaerially erupted products from all ERZ eruptions (Pauahi crater, Episode 1 and 12 of Mauna Ulu) and seismic SWRZ eruptions (1974; ~1700; Fig. 1). Spatter from Mauna Iki on the volcanic SWRZ also contains a significant number of distorted grains. In contrast, erupted products from four intra-caldera summit eruptions and the 1971 volcanic SWRZ eruption contain very few distorted grains. In almost all eruption products, distorted grains are more abundant in sieved fractions >1 mm[29] (Fig. 1). The 1971 volcanic SWRZ spatter contain very few grains >1 mm (4 in ~150 g), but all are distorted, whereas no grains <1 mm show distortion. Intracaldera summit eruptions (August, 1971, September 1971, July 1974, September 1974) contain no grains >1 mm, but all distorted grains are >840 μm.

**Evaluating the dendritic growth hypothesis.** First, we test the hypothesis that lattice distortions are produced during an early phase of branching dendritic growth, followed by textural re-equilibration during periods of reduced undercooling[5]. Due to a change in olivine-melt partitioning during rapid dendritic growth, high concentrations of phosphorous (P) are incorporated into the crystal structure[39]. While the external morphology of dendritic crystals evolves during subsequent textural re-equilibration to obscure this early growth phase, slow diffusion rates preserve internal P enrichments[40]. The presence of P-rich zones passing across subgrain boundaries would be one of the main lines of evidence supporting the dendritic growth hypothesis[5,9]. However, wavelength-dispersive spectroscopy (WDS) maps collected by electron microprobe reveal homogeneous cores within almost all distorted grains (Fig. 2). Similar findings have been reported for olivines from Kīlauea Iki[27]. One P-rich zone terminates against a subgrain boundary (Fig. 2d), suggesting that P enrichments and subgrains did not form during the common process of dendritic growth. Instead, the rapid growth episode generating the P zoning may have preceded the distortion of the crystal lattice, perhaps separated by a dissolution

episode (c.f., ref. [5]). Thus, while some olivines at Kīlauea may have experienced an early dendritic growth phase, this process appears unrelated to the production of lattice distortions.

Furthermore, if distorted olivines represent texturally-matured dendrites, the crystallographic signatures of dendritic growth and lattice distortion should be identical. We examine low-angle subgrain boundaries in 137 distorted olivine crystals from Kīlauea, and compare them to the crystallographic signature of dendritic branching in 61 olivine dendrites from West Greenland[33]. Most branching dendritic buds in the West Greenland olivines show identical crystallographic orientations to the primary crystal; merging of such buds cannot produce lattice distortions. The 125 misorientated crystal buds (Fig. 3a) show a bimodal distribution of misorientation axes, with a strong maximum around [010][41] and a weaker maximum around [100] (Fig. 3c)[34,41]. Misorientation angles are predominantly <20°, but extend up to 90.5° (Fig. 3d)[34]. In contrast, lattice distortions in Kīlauean olivines show misorientation axes predominantly between [010] and [001] (Fig. 4c), and small (<5°) misorientation angles. The boundary between dendritic buds and the host crystal display [010] and [001] weighted Burgers vector directions (WBVDs) (Fig. 3b), while lattice distortions show predominantly [100] WBVDs, with some [001] vectors (Fig. 4b). Finally, the interior of dendritic olivines display no low-angle boundaries with [100] or [001] Burgers vectors.

These crystallographic differences imply that lattice distortions are not produced by textural maturation of olivine dendrites. Instead, the misorientation axes and WBVDs in distorted olivines are indicative of slip systems similar to those invoked to cause CPOs in olivine deformation experiments at conditions resembling Kīlauea's plumbing system[42]. Crucially, in contrast to olivine dendrites, Kīlauean olivines exhibit very few [010] WBVDs (consistent with experimental observations that slip on [010] is unusual due to the long length of the *b* lattice vector[43]). Thus, we propose that lattice distortions represent true deformation of the crystal lattice, rather than growth features.

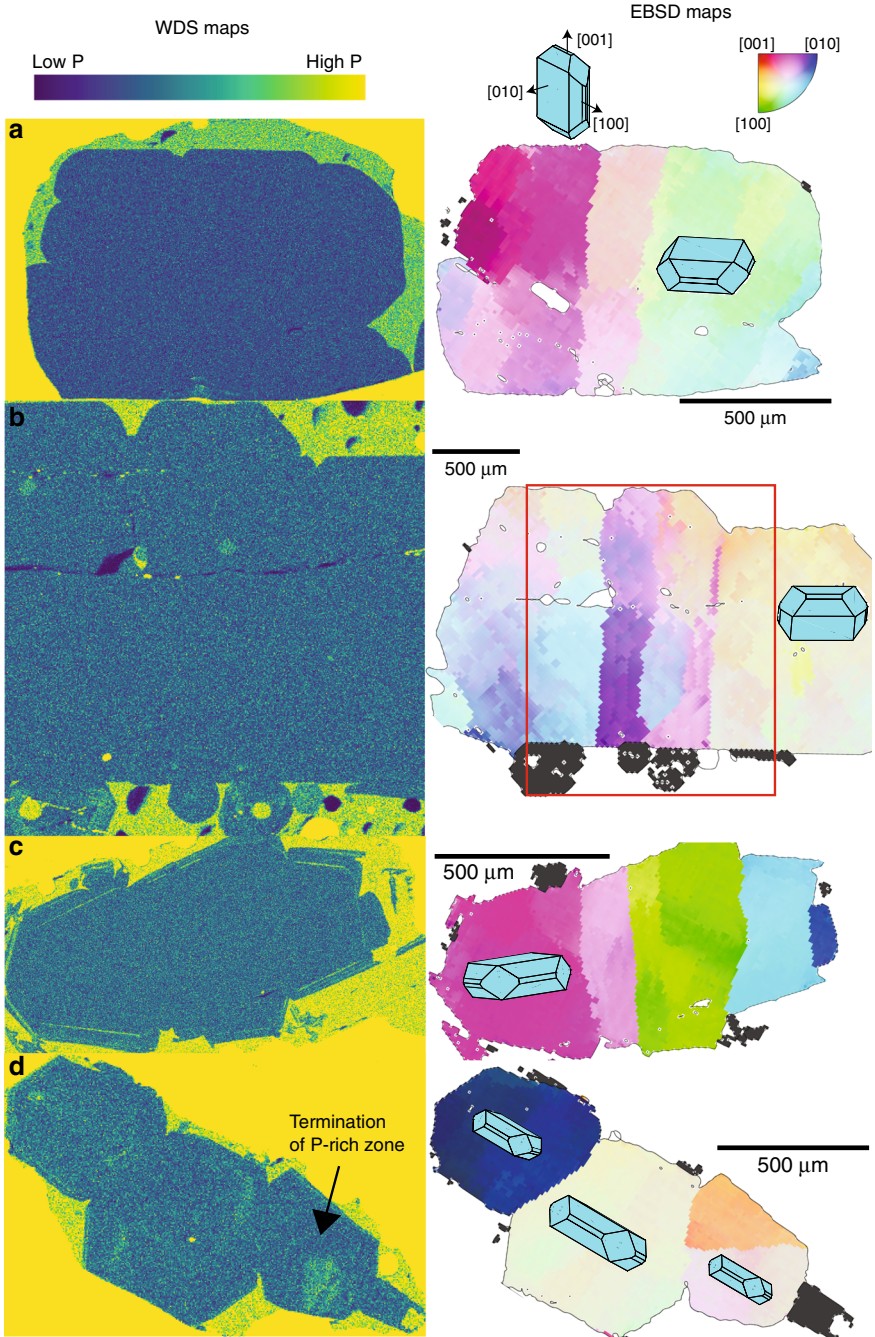

**Fig. 2 Relationships between minor element enrichments and lattice distortions.** Wavelength dispersive spectrometry (WDS) maps of P enrichment compared to electron backscatter diffraction (EBSD) maps (scale for both images shown on EBSD map). **a–c** Grains with internal lattice distortions (revealed by stripes of different colors in EBSD maps) have homogeneous concentrations of P in their cores (see WDS maps). Observed phosphorus enrichments in the rim of the crystal shown in **c** likely mark a period of rapid growth upon eruption. **d** Internal P enrichments are truncated against a subgrain boundary. WDS elemental maps are colored by the number of counts per pixel (in ImageJ). EBSD maps are colored using the inverse pole figure (IPF) key. A color reference direction is chosen such that the mean orientation of the central grain is colored white. The coloring denotes the misorientation axis and angle of each pixel relative to this reference direction. To emphasize small misorientations, the IPF key has been shrunk so that all the color variation is distributed across misorientation angles of 0–3°. For example, pixels misoriented about [010] by 3° from the mean orientation are colored dark blue (see the color scale, top right). Pixels misorientated by >3° are colored black. Light blue 3D olivine crystals are superimposed (using the MTEX crystalShape package) to allow visualization of the orientation of the three crystallographic axes ([100], [010], and [001], for space group Pbnm).

**Collisions of crystals.** The indistinguishable major element chemistry of deformed and undeformed grains[6,38], along with euhedral grain outlines, has led to suggestions that deformation results from grain collisions during magma flow through constricted conduits[10,11]. However, deformed grains examined in this study have higher forsterite contents than undeformed grains

(Fig. 5a, b), with statistically significant differences for individual eruptions ($p < 0.01$, Kolmogorov-Smirnov test, Fig. 5c–f). Differences in minor element compositions have also been reported for deformed and undeformed grains erupted at Kīlauea Iki[27]. Finally, deformed olivines in different eruptions have similar forsterite contents, which are significantly out of equilibrium with

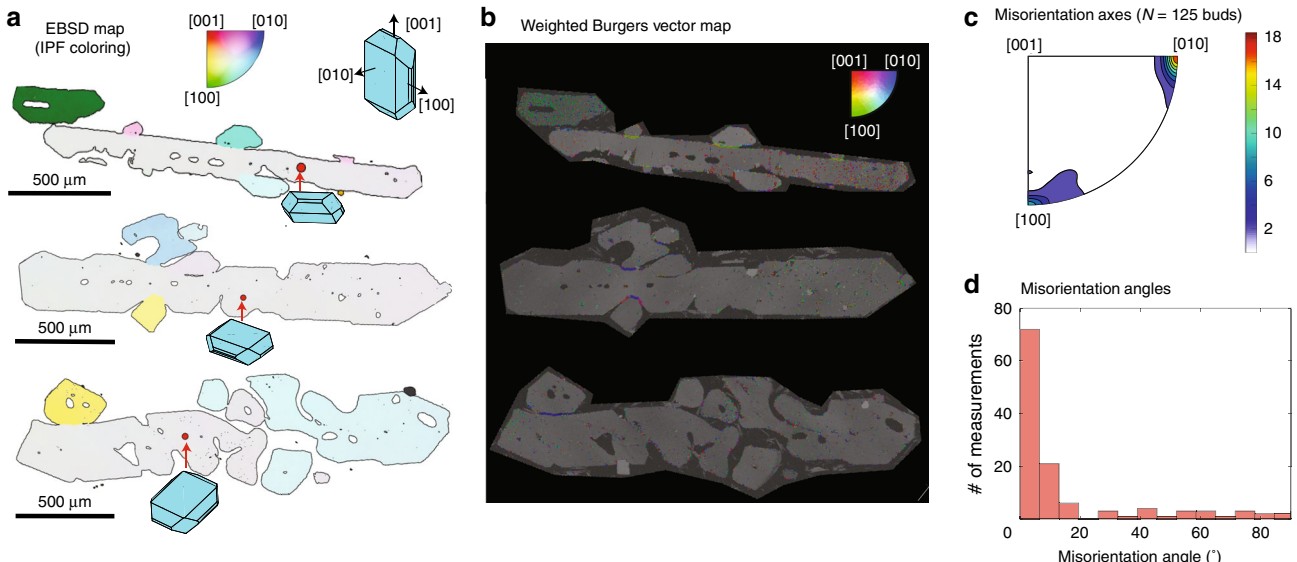

**Fig. 3 Crystallographic signatures of dendritic branching in olivines from West Greenland picrites. a** Electron backscatter diffraction (EBSD) maps colored using the inverse pole figure (IPF) key as in Fig. 2, with the mean orientation of the grain containing the red dot set to white. To emphasize the larger range of misorientation in dendrite branches compared with the distorted olivines in Fig. 2, the IPF color key is adjusted so that pixels with misorientation angles >20° are colored black. 3D olivine crystals are overlain to allow visualization of the orientation of the dendritic olivines with respect to the three crystallographic directions of olivine. **b** Weighted Burgers vector directions (WBVDs) overlain on band contrast maps ([100] = green, [010] = blue, [001] = red]). Scale for both images shown on EBSD map. **c** Misorientation axes for 125 dendritic buds from West Greenland picrites (misorientation angle >1°). The color scale has units of "multiples of uniform distribution". **d** Histogram of misorientation angles between adjacent dendrite buds. Seventy percent of dendritic buds are misoriented from the host crystal by >5° (see ref. [34]).

their matrix glasses. In contrast, undeformed olivines show more chemical variability, and many plot closer to the equilibrium composition (Fig. 5a, b). These observations cannot be reconciled with late stage crystal-crystal collisions, which would result in deformed and undeformed grains having indistinguishable chemistry[27], and the same degree of disequilibrium with co-erupted matrix glasses.

**Generating differential stresses within volcanic plumbing systems.** The majority of deformed olivines at the Earth's surface are derived from mantle peridotites. However, deformed Kīlauean olivines are unlikely to be mantle xenocrysts for two principal reasons. Firstly, almost all deformed Kīlauean olivines have high CaO contents (>0.2 wt%), while the deformed olivines in mantle peridotites and plutonic rocks typically have <0.1 wt% CaO[15,44] (Fig. 6a). Secondly, many deformed Kīlauean olivines have inclusions of melt and euhedral spinels, indicating growth within the volcanic system[8,29]. These observations suggest that these olivines grew, and were subsequently deformed within Kīlauea's plumbing system.

To generate lattice distortions, crystals must be subject to non-hydrostatic stress[6]. Previously, olivine deformation has been attributed to differential stresses exerted within Kīlauea's deep rift zone dunites, during the downward flow of olivine from the summit area to the rift zones, or rapid rift zone extension during south flank earthquakes[15]. However, mechanisms involving deep rift zone processes require specific magma transport paths to bring deformed crystals to the surface. For example, the presence of deformed grains in lavas from the Mauna Ulu eruption have led to suggestions that magma is transported along the basal decollement[8]. However, significant magma transport through the deep rift zones is at odds with geophysical observations, which demonstrate no measurable changes in rift zone opening rates during magma supply surges or rift zone eruptions[18] and

generally supports shallow (<2 km) magma transport[45]. Given the wide spatial and temporal distribution of deformed grains identified in this study, it seems untenable that every eruption received a significant contribution to its olivine crystal cargo from deep rift zone transport paths.

The textural features of deformed olivines are also inconsistent with an origin from low melt fraction dunites (<5% melt)[15]. Olivines derived from such bodies would be expected to show accumulate textures similar to those found in xenoliths erupted at Kīlauea Iki[6] and post-shield Hawaiian lavas[46]. Instead, most distorted grains have subhedral-euhedral outlines with no internal melt films and high angle grain boundaries (Fig. 4). While corrosion followed by overgrowth can produce euhedral crystals from disaggregated dunites[47], this process would preserve internal melt films.

A crucial observation is the fact that deformed olivines have been noted in a variety of basaltic systems. Vinet et al.[4] report their presence in all five of the volcanic centers which they examined in the Andean Southern Volcanic Zone. Deformed olivines have also been noted in the Vaigat formation of West Greenland[33], the Baffin Bay volcanic province[48], Piton de la Fournaise volcano[5], and the Loch Uisg area, Mull[49]. This implies that olivines are deformed by a process which is near-ubiquitous in a wide range of volcanic plumbing systems. We suggest that a plausible mechanism is gravitational loading within melt-rich cumulate piles with ~40% porosity[50] produced by crystal settling[2]. Such an origin accounts for the global distribution of deformed grains, and the absence of accumulate textures at Kīlauea. To investigate this hypothesis further, we assess the location, and conditions under which deformation occurred.

**Assessing the conditions of deformation.** Dominant slip systems in low-angle subgrain boundaries were assessed by calculating misorientation axes, WBVDs, and boundary wall geometries from

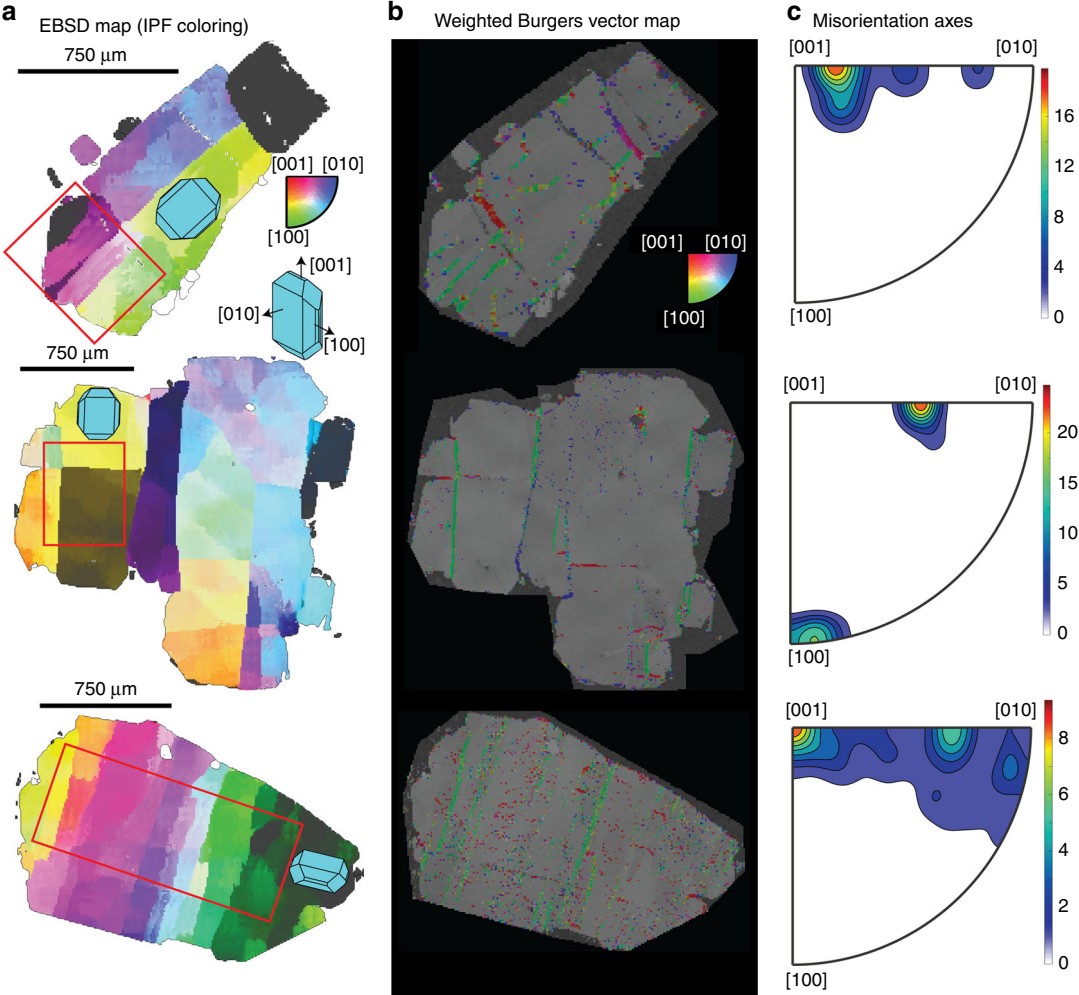

**Fig. 4 Crystallographic signatures of lattice distortions within Kīlauean olivines. a** Electron backscatter diffraction (EBSD) maps colored using the inverse pole figure (IPF) key as in Fig. 2 (scaled to 3°). 3D olivine shapes are overlain to demonstrate the average orientation of the olivine crystal with respect to the three crystallographic axes. **b** Weighted Burgers vector directions (WBVDs) overlain on band contrast maps ([100] = green, [010] = blue, [001] = red). Scale for both images shown on EBSD map. **c** Misorientation axes across subgrain boundaries located within the red box indicated in column **a**. The color scale has units of "multiples of uniform distribution". Subgrain boundaries with [100] WBVDs (green) show misorientation axes between [001] and [010]. Subgrain boundaries with [001] WBVDs (red) show misorientation axes about [100] or [010].

EBSD maps. This approach resolves the ambiguities involved when using misorientation axes alone to determine slip systems[51]. For example, tilt walls consisting of (100)[001] or (001)[100] edge dislocations (described in terms of (slip plane)[Burgers vector]) and twist walls consisting of (010)[100] and (010)[001] screw dislocations all produce misorientation axes about [010]. However, WBVDs (which can be highlighted by color coding of EBSD maps; e.g., Fig. 4b) can distinguish between these three boundary types: (100)[001] tilt walls would show [001] WBVDs (red), (001)[100] tilt walls would show [100] WBVDs (green), and twist walls would show mixes of [100] and [001] WBVDs (green via yellow to red). The geometry of the subgrain boundary provides further discriminatory power. Tilt walls contain the misorientation axis within the boundary plane, while twist wall boundary planes are perpendicular to the misorientation axis (Fig. 7a, b). For simplicity, it was assumed that subgrain boundaries showed pure tilt or twist character. Detailed descriptions of the code and classification criteria used to identify slip systems is provided in the Methods section.

Approximately 83 and 17% of boundaries were classified as tilt and twist walls respectively. These observations are consistent with literature observations that olivine subgrain walls with [100]

WBVDs tend to have edge character[43], and the fact that the majority of WBVDs are [100] or [001] (Figs. 4, 7c), rather than mixed directions (e.g., a WBVD parallel to [101] would appear yellow). Conditions of deformation were deduced from the slip system proportions in the more prevalent tilt walls. Boundaries related to (010)[100], {0kl}[100], and (001)[100] slip comprised ~30, ~44, and ~16% of the total length of tilt boundaries respectively (Fig. 8a). These findings imply that [100] slip systems represent the primary slip systems, with the mosaicity of crystallographic orientation in some olivine grains (Fig. 4) resulting from secondary (100)[001] and (010)[001] slip (~6% and ~3% of boundaries respectively; Fig. 8a).

Experimental work shows that (010)[100] slip, producing A-type fabrics in deformed mantle peridotites, dominates in hot, dry, low stress environments[22]. Slip on (001)[100], producing E-type fabrics, occurs at similar pressure, temperature and stress conditions to A-type fabrics, but is favored by higher water contents (Fig. 8c)[21]. The strengths of these two slip systems are comparable at pressures of 300 MPa, and olivine water contents of ~200 $H/10^6Si_{molar}$[21]. The conditions producing D-type fabrics are debated, they may involve high stresses, low temperatures and low water contents favouring slip on {0kl}[100][20,23].

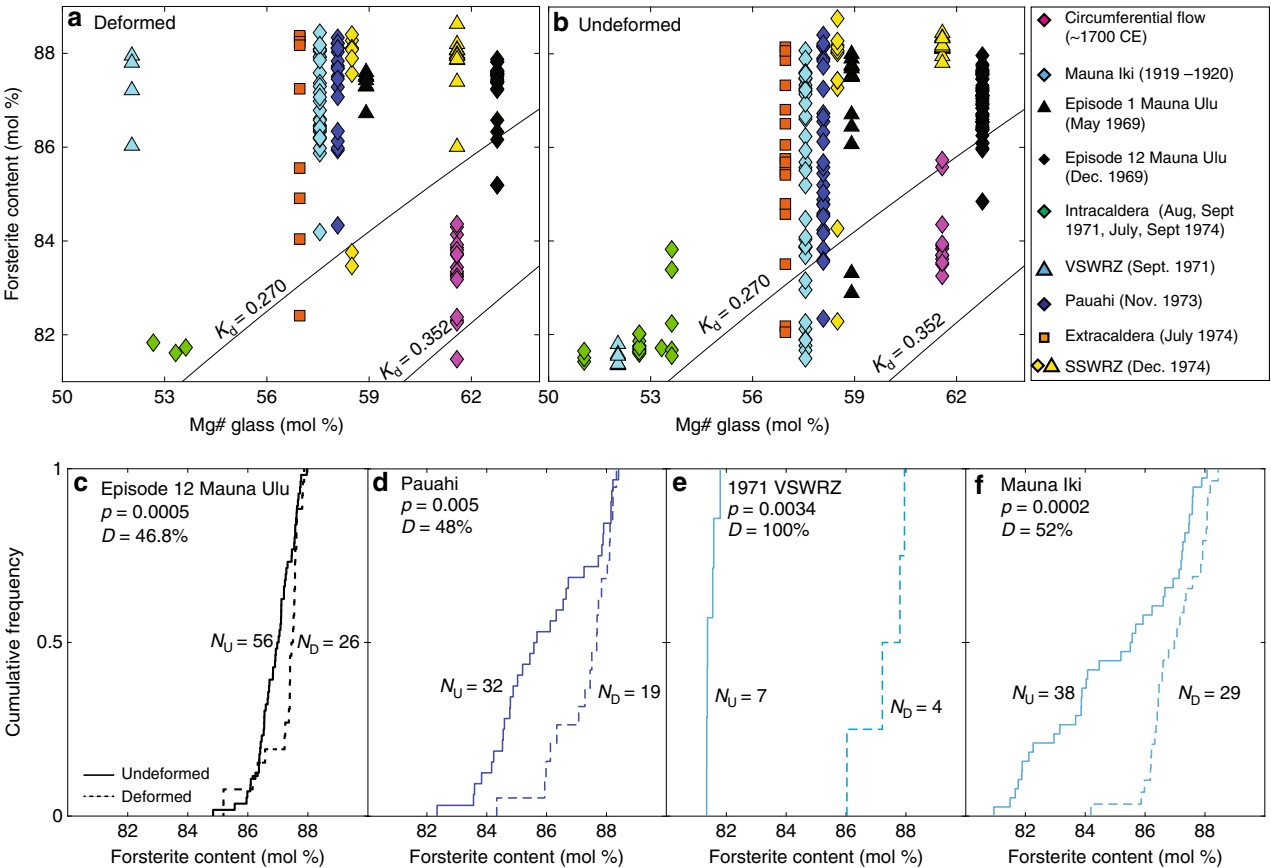

**Fig. 5 Chemical differences between deformed and undeformed Kīlauean olivine grains. a, b** Olivine forsterite contents (mol%) vs. matrix glass Mg# ($Fe^{3+}/Fe_T = 0.15$; ~QFM; see ref. [84]). Olivine-liquid equilibrium lines are shown for $K_D = 0.270–0.352$ (ref. [85, 86]). Both deformed and undeformed grains have higher forsterite contents than the equilibrium olivine composition of their matrix glasses. However, deformed grains show a narrower range of forsterite contents, which lie further from the equilibrium field than undeformed grains. **c–f** Cumulative distribution functions comparing undeformed (solid lines) and deformed (dashed lines) olivines for four eruptions. Deformed grains are more primitive at a statistically significant level ($p < 0.01$) using the two sample Kolmogorov–Smirnov test. The test statistic D (representing the distance between the two distributions) is also shown. SSWRZ, seismic South West Rift Zone; VSWRZ, volcanic South West Rift Zone.

Alternatively, crystallographic signatures indicative of {0kl}[100] slip may be produced when (010)[100] and (001)[100] edge dislocations are present within a single subgrain boundary[36]. Overall, the dominance of slip systems with [100] WBVDs in Kīlauean olivines is indicative of deformation at moderate-high temperatures and low-moderate water contents (Fig. 8a). To further constrain the deformation conditions, and assess the origin of {0kl}[100] slip, we place independent constraints on differential stresses and olivine water contents.

Differential stresses were estimated using subgrain and dislocation density piezometry. As both methods apply to steady state microstructures, they likely represent minimum stress estimates. Subgrain piezometry utilizes the empirical relationship between subgrain-size (d) and differential stress ($\sigma$)[52]:

$$\sigma = \frac{45\mu b}{d} \qquad (1)$$

where $\mu$ is the shear modulus ($65 \times 10^9$ Pa), b is the magnitude of the Burgers vector ($5 \times 10^{-10}$ m) and d is the distance between subgrain boundaries in m.

Subgrain sizes were assessed using an adapted linear intercept method suitable for EBSD maps of individual olivines. The distances between boundaries with [100] WBVDs and misorientations >0.5° were measured perpendicular to the mean boundary trace (Supplementary Fig. 6), rather than along predefined directions in specimen coordinates (c.f. ref. [53]).

Distances were converted into estimates of differential stress using Eq. 1. Subgrain differential stress estimates show a prominent peak at ~3–12 MPa, with 95% of the stresses lying between ~3 and 30 MPa (Fig. 8b).

Dislocation-density piezometry utilizes the relationship between dislocation densities ($\rho$) and differential stress ($\sigma$)[42]:

$$\sigma = \sqrt[s]{\frac{\rho b^2}{B}} \times \mu \qquad (2)$$

where $B = 1.74 \times 10^{-3}$ and $s = 1.37$.

Sakyi et al.[29] report dislocation densities measured using the oxidative decoration technique in a variety of picritic basalts from Kīlauea and Mauna Loa (including some from Mauna Ulu). While they do not report dislocation densities for specific eruptions, their textural descriptions of "blocky olivines" are very similar to our observations of distorted grains. Blocky olivines have dislocation densities of $1–5 \times 10^{10}$ m$^{-2}$, yielding differential stress estimates of ~4–11 MPa. Despite estimates pertaining to different samples, and the large uncertainties associated with each method, stress estimates from subgrain piezometry and dislocation densities are in close agreement (Fig. 8b). Olivine water contents of $H/10^6Si_{molar} = 20–280$ were calculated from $H_2O$ measurements in olivine-hosted melt inclusions from multiple Kīlauean eruptions[54] and olivine-melt partition coefficients ($K_D = 0.0013–0.0021$;[55] orange bar; Fig. 8c).

The low differential stresses estimated from linear intercept distances and dislocation densities are difficult to reconcile with the conventional interpretation that misorientation axes lying

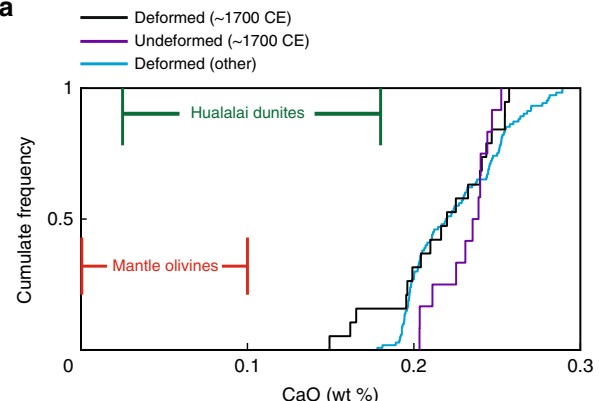

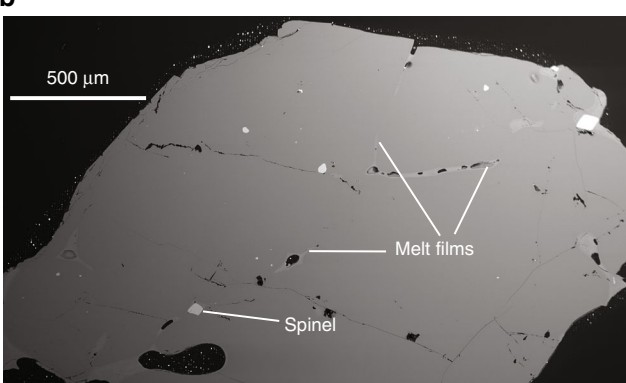

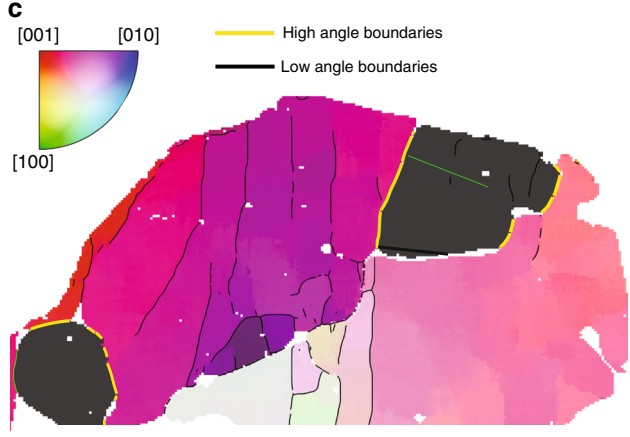

**Fig. 6 Textural and chemical evolution during downslope flow. a** CaO contents of deformed Kīlauean olivines (excluding the ~1700 eruption) plot well outside the field for mantle olivines (see ref. [44]), and have substantially higher CaO contents than olivines from Hualalai dunites (see ref. [15]). Some deformed olivines from the ~1700 eruption have lower CaO contents than undeformed olivines in this sample. In fact, three deformed olivines fall within the compositional field of Hualalai dunites. **b** Backscatter electron image showing prominent melt films along high angle boundaries (yellow lines on **c**) in the interior of a deformed crystal from the ~1700 eruption. **c** Electron backscatter diffraction (EBSD) map colored with the inverse pole figure (IPF) color key scaled to 10° about the mean orientation of the central grain. High angle boundaries (misorientation angle >10°) are colored yellow, and low angle (subgrain) boundaries (misorientation angle <10°) are colored black. Scale for both images shown on the backscatter image.

between [001] and [010] represent slip on the relatively high stress {0kl}[100] system. Instead, our data support the hypothesis that a {0kl}[100]-like misorientation axis distribution is generated by the presence of (010)[100] and (001)[100] edge dislocations within a single subgrain boundary[36]. Slip on (010)[100] dominates at low-moderate stresses and low-moderate water contents, while (001)[100] slip dominates is favored by higher water contents[56] (Fig. 8c). Our independent estimates of olivine water contents and differential stresses lie at this transition where the strengths of (010)[100] and (001)[100] slip are comparable[21]. Thus, it is highly probable that subgrain walls in Kīlauean olivines apparently relating to {0kl}[100] slip are actually made from mixtures of (010)[100] and (001)[100] type dislocations. This reconciles our independent estimates of deformation conditions from subgrain piezometry and the microstructural analysis of slip system proportions.

**Mush pile geometry**. The distribution of differential stress estimates from subgrain and dislocation density piezometry (~3–12 MPa; Fig. 8b) allows assessment of the hypothesis that volcanic olivines are deformed by gravitational loading within melt-rich mush piles. Differential stresses ($\sigma_D$) in mush piles scale as a function of the density difference between crystals and host melts ($\Delta\rho$), and the thickness ($H$):

$$H = \frac{\sigma_D}{g\Delta\rho} \qquad (3)$$

where $g = 9.8$ m/s², and $\Delta\rho = 567$ kg/m³ (using $\rho = 3282$ kg/m³ for Fo$_{86}$ olivine at 1230 °C[57], and $\rho = 2715$ kg/m³ for the interstitial basaltic liquid[54,58]).

Using this simple equation, the measured differential stresses may be generated at the bottom of mush piles ~540–2160 m thick. However, centuries of observations have demonstrated that a near-ubiquitous feature of granular materials is the arrangement of solid particles into force chains, resulting in unequal transmission of load (e.g., corn in silos, stacked fruit, poppy seeds; photoelastic particles)[59–62]. Highly stressed chains of primary members transmit most of the force, while a larger group of secondary members experience less than the mean force, and a few spectator particles experience no force (Fig. 9b)[63]. The settling of dense olivine crystals into mush piles creates a crystal framework with ~40% porosity[50], which constitutes a granular medium. Thus, it is highly probable that mush piles contain force chains. Simulations of mush piles show that the proportion of particles experiencing more than the mean force has an exponential distribution, with forces on certain grains exceeding 6× the mean force[60]. Dynamical phenomena within magma chambers (e.g., magma injection) drive the migration of force chains, resulting in a large proportion of olivines experiencing greater than the mean force[60]. This process could reduce the required mush pile thickness by a third, to ~180–720 m.

More precise constraints on mush pile thicknesses in igneous systems from investigations of deformed volcanic crystals will require several conceptual advances. Numerical simulations of granular mush piles are computationally expensive, despite only considering spherical particles in 2D[60]. In reality, the force applied at contacts between angular crystals will be amplified into very high stresses by small contact areas[64]. Such highly heterogenous stress distributions within individual grains[64,65] complicate piezometric estimates. An increased understanding of how heterogenous stresses within individual grains are translated into subgrain sizes and dislocation densities is needed, alongside discrete-element models accounting for force chain geometry in three dimensions, using realistic crystal shapes. Finally, transitions between a predominantly frictional regime (with force chains) to a lubricating regime may occur within mush piles,

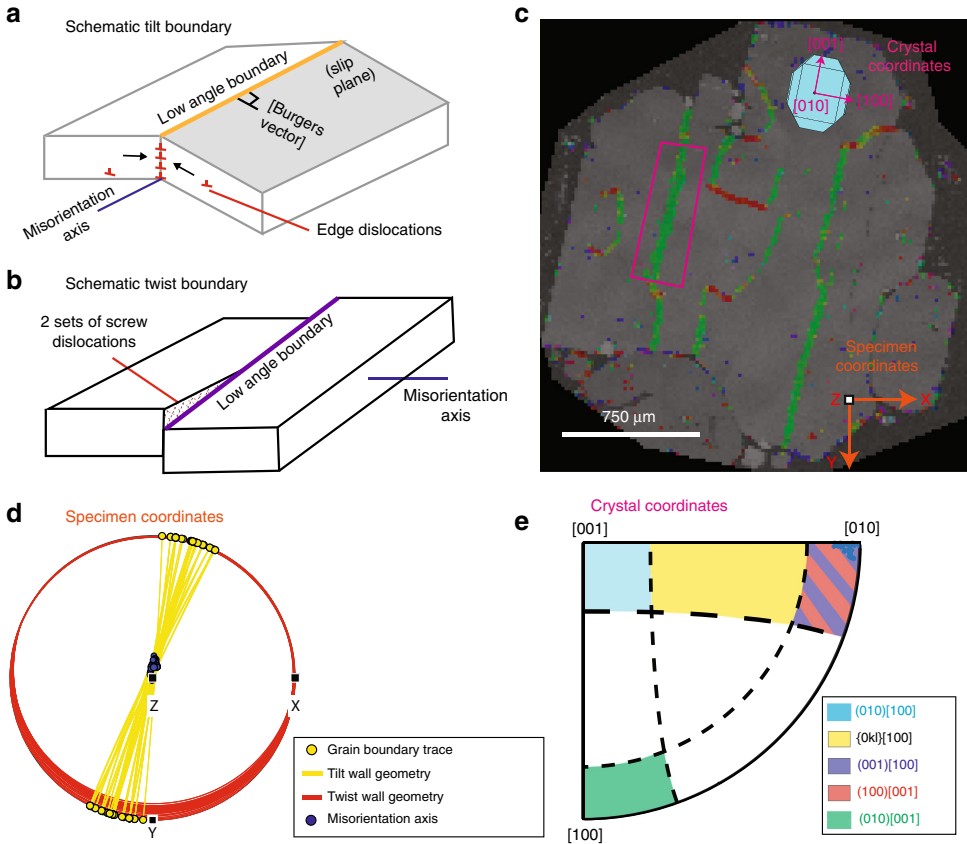

**Fig. 7 Slip system identification method developed for this study. a** A schematic diagram of a tilt boundary, where the misorientation axis lies within the plane of the low angle (subgrain) boundary. **b** A schematic diagram of a twist boundary, where the boundary plane is perpendicular to the misorientation axis. Two sets of screw dislocations are shown within the twist wall for illustrative purposes; this is just one possible geometry. **c** Map of weighted Burgers vector directions (WBVDs) in a deformed olivine. Specimen coordinates (orange arrows) represent the x, y, and z directions within the scanning electron microscope. Crystal coordinates (pink arrows) represent the directions of the [100], [010], and [001] axes of the olivine. A 3D olivine crystal is overlain to demonstrate the orientation of the crystal in crystal co-ordinates. Boundaries with [100] WBVDs are almost parallel to the y direction, while those with [001] WBVDs are almost parallel to the x direction in specimen coordinates. A small section of a [100] WBVD boundary is selected for further discussion (pink rectangle). **d** Ideal tilt (yellow) and twist (red) boundaries calculated for each boundary segment in the pink rectangle based on the trace of the subgrain boundary and direction of the misorientation axis (in specimen coordinates). As the calculated twist wall planes show extremely low dips, the MTEX code classifies all segments along this boundary as tilt walls. **e** Schematic diagram showing the misorientation axes in crystal coordinates for tilt boundaries built from dislocations types associated with each of the 5 olivine slip systems (written as (slip plane)[Burgers vector]), allowing 20° leniency. Misorientation axes of the boundary pixels within the pink rectangle plot near [010]. Along with the [100] WBVDs, this indicates that this tilt wall consists of (001)[100] dislocations.

affecting the amount of stress amplification[60]. An alternative research direction would be to perform EBSD on olivines at the base of mush piles of known thickness in extinct volcanic systems (where intrusions outcrop at the surface), to produce a calibration between deformation intensity and mush pile thickness.

**Mush pile location**. The spatial distribution of deformed grains across the Kīlauean edifice provides insights into the location of mush piles. Geophysical and geochemical observations at Kīlauea have led to a model in which magma is stored at two distinct depths[66–68]. We find the largest proportion of deformed olivines in eruptions derived from the deeper South Caldera (SC) reservoir (based on Pb isotopes)[67,68]. In contrast, there are few deformed grains in intracaldera and volcanic SWRZ eruptions which tap the upper, more evolved Halema'uma'u reservoir. The different sources of these eruptions is also evident from the higher Mg# of co-erupted glasses in eruptions containing abundant deformed grains (Fig. 5a, b), consistent with the storage of more primitive melts in the deeper SC reservoir[67,69]. Interestingly, unlike the 1971 SWRZ eruption, the 1919–1920 Mauna Iki

eruption contains abundant deformed grains (both eruptions occurred on the volcanic SWRZ). This supports the hypothesis that the Mauna Iki eruption tapped magma from multiple reservoirs based on the decoupling of E–W and N–S tilt components[37].

Crucially, the commonality between eruptions containing deformed grains is that a significant proportion of their magma supply (and thus crystal cargo) was derived from the SC reservoir. An origin for deformed grains within mush piles at the base of this reservoir is a simpler explanation for their abundance in a wide variety of subaerial eruptions than invoking magma transport paths along the basal decollement[8]. The narrow range of forsterite contents in deformed grains in our study (~$Fo_{84-88}$; Fig. 5) may result from diffusive re-equilibration of Mg and Fe during prolonged mush pile storage[2,70]. The scavenging of deformed grains from mush piles just before eruption also explains the high degree of olivine-liquid disequilibrium between deformed grains and their host melts[70]. As the Halema'uma'u reservoir is thought to be recharged from the SC reservoir[67], entrainment of a small number of deformed grains during

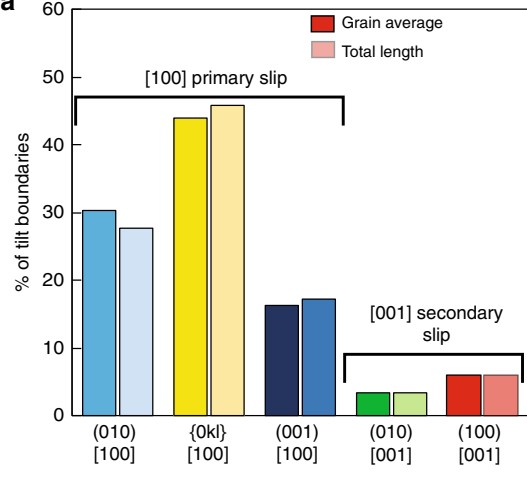

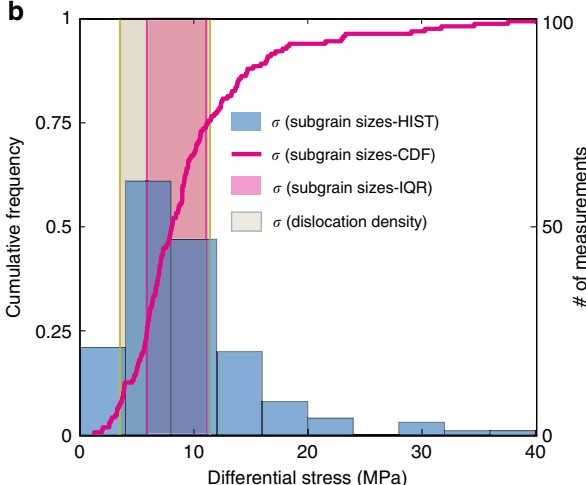

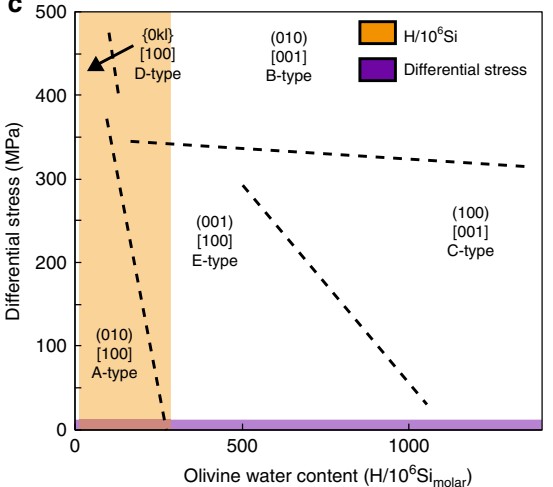

**Fig. 8 Quantifying the conditions of deformation. a** Proportions of each slip system calculated by averaging the proportion of each slip system per grain (darker colors; giving equal weight to all grains regardless of their size), and by summing the total length of each boundary type in 62 deformed grains (lighter colors). All deformed grains contain [100] weighted Burgers vector directions (WBVDs), suggesting that this is the primary slip direction. Only a subset of grains contain low angle boundaries with [001] WBVDs, suggesting that this is a secondary slip direction. **b** Differential stresses calculated using subgrain piezometry (see ref. [52]) applied to linear intercept distances clearly overlap with those estimated from dislocation density piezometry (see ref. [42]) on published dislocation densities (see ref. [29]). Differential stress estimates are shown as a cumulative density function (CDF-magenta) with the interquartile range (IQR-light pink), and as a histogram (HIST-darker blue). **c** Olivine regime diagram (see ref. [56]) for $T = 1470-1570$ K with subgrain stress estimates (purple bar) and olivine water contents (orange bar) superimposed.

geochemical constraints at Kīlauea. Primary melts with 16.5 wt% MgO enter a cylindrical sill at a rate of 0.1 km³/year[71] (Fig. 10a). Within the sill, olivine crystals form and settle into cumulate piles (~40% porosity), causing the remaining liquid to evolve to lower MgO contents. This liquid leaves the reservoir, and is either erupted, or stored within the deep rift zones[71]. Fractionation from primary liquids to the average whole rock MgO contents of erupted lavas (~10 wt% MgO) requires removal of ~14 vol.% olivine cumulate[15]. The thickness of the cumulate pile in the sill is a play off between reservoir radius and accumulation time (Fig. 10b). Geodetic observations of surface displacements by interferometric synthetic aperture radar (InSAR) can be modeled as a magma chamber represented by penny-shaped cracks with radii of ~2–4 km[72]. This geometry permits accumulation of ~180–720 m of mush with ~40% porosity in just a few centuries (Fig. 10). This is consistent with the diverse range of trace element ratios within melt inclusions in any given eruption, which indicate minimum mush pile residence times of > 170 years[70]. Thus, our hypothesis that deformed grains are generated by gravitational loading within mush piles accumulating within the SC reservoir is supported by available geometric and temporal constraints. Under this framework, deformed olivines are not archives of processes occurring within Kīlauea's deep rift zones. However, the absence of petrological evidence for dunitic bodies does not imply their absence (c.f. ref. [5]). Many deep processes within igneous systems are not manifested in erupted products[73], but still play a crucial role in the evolution of volcanic systems.

The temporal evolution of mush piles at the base of the deeper SC reservoir is uncertain. To resolve space problems, olivine must eventually migrate downslope to form the seismically imaged dunitic bodies[15] (Fig. 9a). We suggest that the amount of interstitial melt gradually reduces during this process, leading to the generation of true adcumulate textures, with internal melt films and high angle grain boundaries. Additional stress imposed during downslope flow accounts for the higher differential stresses estimated from subgrain piezometry in dunitic xenoliths from Hualalai (40 ± 20 MPa vs. 3–12 MPa)[46]. Deformed olivines from the ~1700 eruption may record the first stages of this process; they contain prominent low and high angle boundaries (misorientations <10° and >10° respectively; Fig. 6b, c). The latter are decorated with thin films of melt (Fig. 6b), indicating derivation from low melt fraction bodies. These olivines also show trails of secondary fluid inclusions, similar to those observed in Hawaiian xenoliths[15]. Finally, these olivines have substantially lower forsterite contents than other deformed olivines (~Fo$_{82-84}$ vs. ~Fo$_{84-88}$; Fig. 5a, b), and several have CaO contents overlapping with values measured in Hualalai

recharge events can account for their occasional occurrence in intracaldera eruptions, and the more extreme difference in forsterite contents between deformed and undeformed grains in these eruptions (Fig. 5e).

To further assess the hypothesis that deformed grains form in mush piles accumulating at the base of the SC reservoir, we explore the timescales required to generate the mush piles thicknesses indicated by subgrain piezometry (~180–720 m) using a simple box model informed by available geophysical and

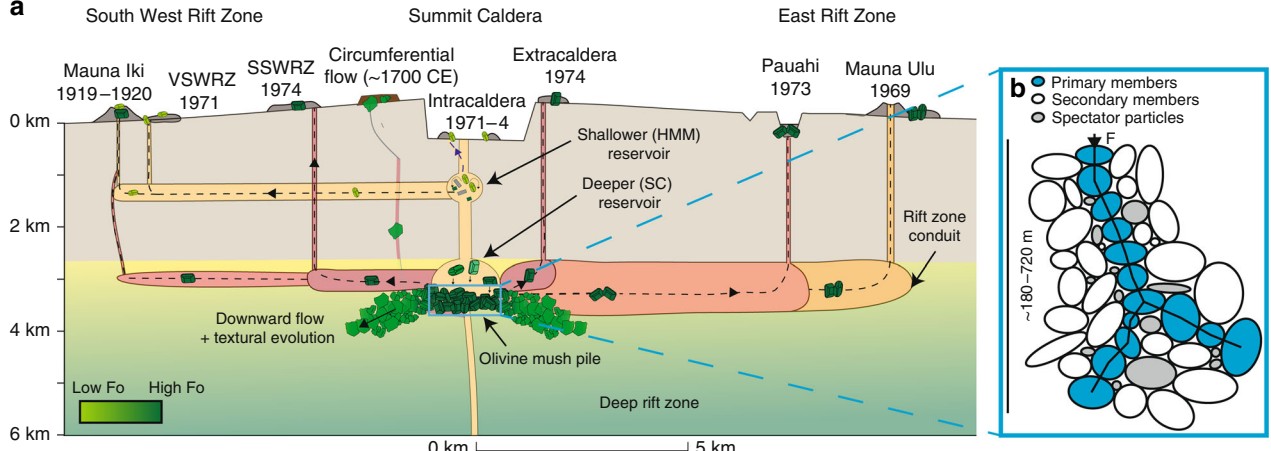

**Fig. 9 Schematic cross section along the magmatic plumbing system of Kīlauea showing the formation of deformed olivines. a** Olivine crystals in intracaldera eruptions thought to tap the shallower reservoir based on Pb isotope compositions (see refs. [67, 68]) are largely undeformed. These crystals have evolved forsterite contents, close to the equilibrium composition of their matrix glasses. Extracaldera and rift eruptions, thought to tap the deeper SC reservoir based on Pb isotopes (see refs. [67, 68]), entrain deformed crystals from melt-rich olivine mush piles at the base of the SC reservoir just prior to eruption. The Mauna Iki eruption taps a mixed crystal cargo, from the upper and lower reservoir (see ref. [37]). The ~1700 eruption and the Kīlauea Iki eruption (see ref. [6]) may tap olivine bodies that have begun their descent into the deep rift zones. **b** Schematic diagram of force chains within granular olivine mush piles (adapted from the ref. [60]). Primary members can experience forces ~6× the mean force, secondary members experience less than the mean force, while spectator particles experience no force.

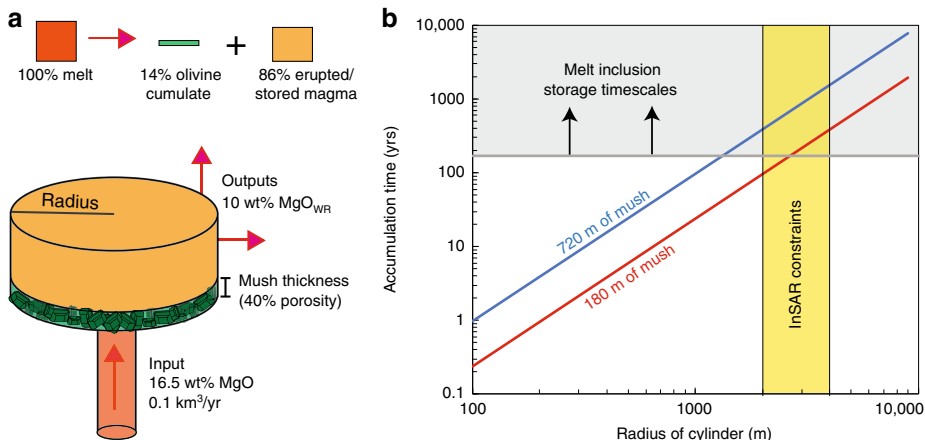

**Fig. 10 Estimating mush pile accumulation rates using a simple box model. a** A cylinder is supplied with primary melt (16.5 wt% MgO) at a rate of 0.1 km³/year (see ref. [71]). Fractionation of these primary melts to the average whole-rock MgO contents of erupted lavas (~10 wt%) requires fractionation of 14 vol.% olivine crystals (ref. [15]). **b** The play off between cylinder radius and accumulation time is shown for mush piles with thicknesses of 180 and 720 m (assuming 40% porosity). Available constraints on reservoir radius from interferometric synthetic aperture radar (InSAR) are shown in yellow (see ref. [72]). Melt inclusion trace element diversity at Kīlauea indicates that crystals are stored in mush piles for >170 years (see ref. [70]; gray box). These geometric and temporal constraints bracket the range of mush thicknesses estimated in this study.

dunites[46] (Fig. 6a). These chemical differences may reflect cooling during transport away from the main magma storage region at Kīlauea. Texturally similar olivines have also been identified in the Kīlauea Iki eruption[6]. Eruptions sampling these more consolidated dunitic bodies are clearly uncommon, and may be restricted to those ascending along infrequently utilized pathways to the surface[6].

**Microstructural analysis constraints mush processes.** A quantitative examination of microstructures in distorted olivines erupted at Kīlauea Volcano by EBSD reveals striking similarities to deformed mantle peridotites, but significant differences to the crystallographic signatures of dendritic growth. This suggests that lattice distortions record the application of differential stresses at

high temperatures within the magmatic plumbing system, rather than rapid crystal growth. Previous petrological work has concluded that differential stresses are applied during the creep of dunitic bodies within Kīlauea's rift zones. Crucially, this has fueled suggestions that significant quantities of magma bypass the summit reservoir and travel along the deep rift zones, despite the paucity of geophysical evidence for this process. In contrast, we suggest that olivine crystals are deformed in gravitationally-loaded olivine mush piles at the base of the deeper South Caldera reservoir, explaining the spatial distribution of grains at Kīlauea, and the absence of adcumulate textures. Moreover, the mechanism we propose here accounts for the occurrence of deformed grains in a wide variety of basaltic systems worldwide, which lack deep rift zones.

Application of piezometers developed for mantle peridotites reveals that deformed olivines experienced differential stresses of ~3–12 MPa. Assuming that olivine mush piles behave as granular materials, and form force chains, the vertical extent of mush piles at Kīlauea can be estimated (~180–720 m). Based on available constraints on the magma supply rate, and the geometry of Kīlauea's summit reservoir, these thicknesses accumulate in a few centuries.

Overall, microstructural investigations of erupted olivine crystals by EBSD generates rich datasets which provide quantitative insights into crystal storage within mush piles. Under the increasingly prevalent view that crustal magmatic systems are mush-dominated[74], constraining the geometry and dynamics of crystal storage regions is crucial to further our understanding of magmatic plumbing systems. The presence of deformed olivines in many different volcanic settings highlights the global applicability of the methods developed in this study. Furthermore, assessments of deformation conditions using EBSD need not be restricted to olivine-bearing samples. Microstructural fabric types in natural and experimental samples have been established for a wide variety of igneous phases (e.g., diopside[75], plagioclase[76], and hornblende[77]), extending the applicability of our study to more evolved igneous systems.

## Methods

**Sample preparation.** The vesicular nature of the basaltic scoria from Kīlauea Volcano was not amenable to thin section production. Additionally, due to the high modal proportion of vesicles, few olivines were found within each thin section. Instead, tephra and lava samples were jaw crushed and sieved into four size fractions (>1 mm, 1 mm–840 μm, 840–250 μm, and <250 μm). Olivine crystals were picked and mounted in epoxy stubs based on their crystal size (allowing stubs to be variably ground down such that the cores of crystals were exposed). A sample of a West Greenland picrite, and one Kīlauean epoxy grain mount were made into thin sections to allow comparisons between optical and EBSD observations (Supplementary Figs. 1–5). Epoxy mounts and thin sections were polished with progressively finer silicon pastes, then colloidal silica using a VibroMet 2 Buehler Vibratory Polisher.

**EBSD analysis.** EBSD data was collected on a FEI Quanta 650FEG SEM equipped with a Bruker e-Flash HR EBSD detector in the Department of Earth Sciences, University of Cambridge. Data collection and indexing was performed with Bruker QUANTAX CrystaAlign. The step size of each acquisition was varied depending on the size of the aggregate such that each map took ~15–20 min to collect (see Supplementary Table 2 for additional acquisition parameters). EBSD data clean up and post-processing was performed in MTEX V5.0.4[78], an open-source toolset for MATLAB. Single misindexed pixels were removed, and data was denoised using a 5 neighbor Kuwahara filter[31].

**EPMA analysis.** Electron microprobe spot analyzes and WDS maps of olivine were performed using a Cameca SX100 EPMA in the Department of Earth Sciences, University of Cambridge. Olivine spot analyzes were conducted in several analytical sessions, with slight variations in analytical conditions. Count times and calibration materials, alongside estimates of precision and accuracy are shown in Supplementary Table 3. Precision and accuracy were calculated from repeated measurements of a San Carlos Olivine secondary standard. Within each run, deformed and undeformed grains from each eruption were analyzed in roughly equal proportions. Thus, chemical differences between deformed and undeformed grains cannot be attributed to analytical variation.

Matrix glasses from ten eruptions were analyzed in a single analytical session at 15 kV, 6 nA, using a 5 μm defocused beam (samples KL0908, KL0909, KL0910, KL0914, KL0916, KL0917, KL0919, KL0920, KL0922, and KL0931). Glass compositions for samples KL0921, KL0930, and KL0924 were taken from Sides et al. (see ref. [54]; codes 1700g, 1920g, and 1974J2 respectively). Count times and calibration materials, alongside estimates of precision and accuracy are shown in Supplementary Tables 4 and 5. Precision and accuracy were calculated from repeated measurements on VG2 and Basalt_113716-1.

WDS maps of phosphorus and other elements were collected at 15 kV, 200 nA, with a spot size of 1 μm (see Supplementary Table 6 for details). The pixel size was varied between 3–6 μm depending on the size of the olivine, with a dwell time of 400 ms. The total acquisition time for each map varied between 3–17 h.

**Slip system identification.** The classification of a grain as distorted or undistorted is somewhat arbitrary, particularly based on optical observations[28,29]. Here, the presence of distortion was quantitatively assessed by identifying subgrain boundaries using the calcGrains MTEX function, with a threshold angle of 0.5°. Olivines

were classified as distorted if at least 1 low angle boundary stretching across a third of the crystal was identified. The use of smaller threshold angles than 0.5° would result in imprecise estimates of misorientation axes and angles, due to the finite angular resolution of conventional EBSD[79,80]. Furthermore, utilization of smaller threshold angles can result in inaccurate stress estimates from microstructural methods such as subgrain piezometry[53].

The same detector resolution was used for all EBSD acquisitions, and pixel spacings were significantly smaller than the microstructure features of interest (3–10 μm vs. ~100 μm), resulting in a consistent definition of a distorted crystal. The shape of the boundary and relation to external morphological features in BSE images were visually assessed to ensure that grains containing low angle boundaries resulting from grain aggregation[34] were not considered further (e.g., the boundaries between euhedral grains in Fig. 2d).

To allow comparison of our EBSD data with previous studies utilizing optical microscopy at Kīlauea, EBSD maps for ~20 grains within a ~40 μm thick section were compared to optical observations. Low angle boundaries with misorientations <1° were extremely difficult to see optically, even with a high-spec optical microscope (Zeiss Axioscope A1; Supplementary Figs. 1–5). Additionally, the ability to optically resolve subgrain boundaries depended on the sharpness of the misorientation contrast, and the orientation of the crystal within the thin section. In some instances, subgrain boundaries with misorientations up to 2.5° were invisible. Based on the small subset of grains examined here, previous optical studies may have underestimated the proportion of distorted grains by up to 45%. This is similar to the conclusions of Sakyi et al.[29] who observe dislocation textures in optically undeformed grains.

We quantify the dislocations present (and thus the dominant slip systems) within distorted olivine crystals using two metrics that can be obtained from EBSD data; the misorientation across a boundary, and the WBVD. The misorientation describes the axis-angle pair of the coordinate transform from the crystallographic orientation on one side of the boundary to that across the boundary. Misorientations are stated in terms of a misorientation axis and the minimum angle of rotation about this axis[81]. For example, a misorientation of 3° about [010] shows that the crystallographic directions of two adjacent subgrains can be superimposed by a rotation of 3° about the olivine b axis.

The WBVD quantifies the weighted mean of the Burgers vectors of all types of geometrically necessary dislocation in a local region, assessed using 2D orientation gradients in EBSD maps (see ref. [82] for more detail). This is also expressed in terms of a crystallographic direction (e.g., [100]). The WBVD depends on the dislocation density for each type, and is weighted towards dislocation lines at a high angle to the map. For discrete boundaries, the apparent dislocation density and hence the WBV magnitude depends on pixel spacing, but the WBV direction does not, and provides sufficient crystallographic information to assess slip system occurrence. Specifically, the WBVD samples the Burgers vectors of dislocations in low-angle boundaries and cannot introduce any illusory directions which are not present in the actual microstructure.

Crucially, EBSD allow assessment of the misorientation and WBVD for lots of neighbouring pixels along a given low angle boundary, and large numbers of individual low angle boundaries in relatively short acquisition times (15–20 min per grain) and processing times (~2 min per grain). The large datasets generated using these methods are crucial for the precise characterization of microstructures in volcanic olivines which show only subtle distortions.

Subgrain boundaries within individual EBSD maps were classified using new code built in MTEX 5.0.4[78], incorporating WBVD calculations from CrystalScape[82]. Firstly, misorientation axes were calculated for boundary segments with misorientation angles between 0.5–10°. Boundaries were classified as twist walls if they met the following criteria: (a) angle between the misorientation axis and the z direction in specimen coordinates >15° and (b) angle between the misorientation axis and the grain boundary trace in specimen co-ordinates >75°. Criteria (a) accounts for the fact that boundaries with low dips are unlikely to be observed[82]. Crucially, this rules out misidentification of a tilt boundary orientated such that the misorientation axis is near vertical as a twist wall. The second criteria accounts for the theoretical geometry of a twist wall (Fig. 7b), where the misorientation axis is perpendicular to the boundary trace (allowing for 15° leniency). Boundaries were classified as tilt walls if: (c) the boundary was not classified as a twist wall and (d) the normal to the plane containing the grain boundary trace and misorientation axis (i.e., the normal to the tilt boundary plane) in specimen coordinates was >15° from the z direction (discarding boundaries with low dips). For example, boundary pixels within the pink rectangle (Fig. 7c) meet (b) but not (a), as the misorientation axes are nearly vertical (blue dots; Fig. 7d). Thus, boundary pixels were classified as a tilt walls, as they met condition (c) and condition (d). This tilt wall is orientated with a near vertical boundary plane and misorientation axis.

As this method classified only ~17% of boundaries as twist walls, slip system proportions were quantitatively assessed in the more prevalent (and more geometrically simple) tilt walls. Within a single grain, tilt boundaries with a given WBVD displayed a narrow range of boundary traces (Fig. 7c), allowing isolation of tilt boundaries with [100] and [001] WBVD. For boundaries displaying each WBVD, the direction of the misorientation axis for each boundary segment was compared to the misorientation axis (in crystal coordinates) expected for each slip system (Fig. 7e). Slip system proportions were calculated in two ways; by summing the total length of each boundary type, and by calculating the proportion of each boundary type in a single grain, and averaging these values (N = 62 grains, Fig. 8a).

**Linear intercept method**. Linear intercepts are traditionally measured along predetermined directions (e.g., horizontal and vertical) within entire thin sections. We adapt this method for use on EBSD maps of individual deformed olivine crystals. Instead of measuring along arbitrary directions, we measure distances perpendicular to low angle boundaries with [100] WBVDs within the interior of the crystal. Distances between the outermost subgrain boundary and the edges of the crystal were not used, as this distance depends on the amount of crystal preserved during jaw crushing, and the extent of crystal growth following their scavenging from mush piles (ref. [70]; see Supplementary Fig. 6). It was decided that sectioning effects were negligible compared to the uncertainty stemming from different parameterizations of the relationship between subgrain size and differential stresses[53].

## Data availability

All compositional data obtained in this study are included as Supplementary Data tables.

## Code availability

The Matlab scripts used to classify olivine slip systems (run in MTEX 5.0.4) have been uploaded into the Supplementary Information with an example.ctf file of the distorted grain shown in the middle row of Fig. 4.

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

## Acknowledgements

We acknowledge Zoja Vukmanovic, Lars Hansen, and Chris Gregson for very helpful discussions about identifying slip systems. Isobel Sides (funded by a Natural Environmental Research Council [NERC] studentship) and Don Swanson collected the Kīlauea samples used in this study, and Lotte Larsen is thanked for providing the Greenland picrite samples. Giulio Lampronti and Iris Buisman are thanked for help collecting EBSD and EPMA data. P.W. is funded by NERC DTP studentship NE/L002507/1.

## Author contributions

P.W. conceived the project with help from M.E. and J.M. P.W. prepared the samples, collected and processed the EBSD and EPMA data and wrote the MTEX code. J.W. provided the WBVD scripts and guidance interpreting the EBSD data. P.W. wrote the manuscript with help from M.E., J.M., and J.W.

## Competing interests

The authors declare no competing interests.
