## [Peer Review File · Nature Communications]

Reviewers' comments:

Reviewer #1 (Remarks to the Author):

Wieser et al. have performed a careful study of strain in olivine in Kilauea eruptives (and Greenland for comparison) and they have used that data together with several estimated parameters to perform two piezometer calculations which allows them to interpret the location of formation for their samples (using pressure, temperature and water content estimates/assumptions), and furthermore to estimate the thickness of the mush pile. They have done a considerable amount of work in this contribution, and the data appear technically sound, and make an important contribution to the scientific community. The data collection methods are not new, but the broad-ranging interpretation of their data advances our understanding about the origins of deformed olivine. However, there are some uncertainties in the interpretation, and the certain areas of the paper require clarification, as outlined below.

The authors have done a good job of recognizing previous contributions to the literature. The authors have done an excellent job of going back to the source for the techniques (e.g. Donaldson 1976 olivine morphology; Toriumi 1979 piezometry; and Prior 1999 EBSD & OC imaging). But what about the work by J.-C. J. Mercier, who wrote "Chapter 19. Olivine and Pyroxenes" in the 1985 text book by Wenk, Ed. "Preferred Orientation in Deformed Metal and Rocks. An introduction to Modern Texture Analysis" (1985), which shows textures, slip systems, and pole figures for olivine in various settings, including piezometry.

I think it would be useful to add optical photomicrographs where possible to Figures 2, 4, 6, 7, to compare with the EBSD images, to emphasize the improvement in quality of the data by EBSD. On Lines 76-77, the authors state that this work "offers significant improvements in spatial and angular resolution compared to optical observations". On lines 606-607, they claim that using optical observations, misorientation boundaries < 1 degree (up to 2.5 degrees) were invisible. But the MTEX lower threshold of 0.5 degrees is arbitrary, and only one boundary needs to be identified (line 595).

I would like to see more clarity in the pressure obtained by their observations. Their piezometry calculation suggests relatively low pressure (~3-12 MPa), allowing them to discount rift zone origin, but there is uncertainty over interpretation of one set of slip directions as either {0kl}[100] slip (high stress, low T, low H₂O – line 266) or a combination of (010)[100] and (001)[100] - line 268 (indicating deformation at mod-high T, low-mod H₂O – lines 272-273) . These two sets of observations seem to lead to a circular argument:

Lines 302-308 – "The apparent presence of {0kl}[100] slip is at odds with our differential stress and water content estimates (Fig. 8c), supporting our earlier suggestion that subgrain walls apparently relating to {0kl}[100] slip are in fact made from mixtures of (010)[100] and (001)[100] type dislocations. Our estimates for water contents and differential stress, which indicate conditions where the strengths of (010)[100] and (001)[100] slip are comparable, are then entirely consistent with our microstructural analysis of slip system proportions."

If the active slip system was assumed to be {0kl}[100], would this provide a higher calculated pressure using dislocation piezometry? It would be useful to be able to definitively test the existence of {0kl}[100] slip in order to clarify their argument.

The force chain argument also seems like a stretch here. The problem with invoking this theory here is that there is no evidence in the data measured to test support for this proposed mechanism. Only single grains are observed (not a dunitic texture) so the proposed force chain mechanism can not be confirmed by observations within this context. I like the authors' suggestion (Lines 354-356) "An alternative research direction would be to perform EBSD on olivines at the base of mush piles of known thickness (e.g. exposed sills) to produce a calibration between deformation intensity and mush pile thickness." but apparently, observations of the base of mush piles are uncommon (line 401).

Clarity of the writing is also an issue. I realize that Nature Communications caters to experts in the field, but the level of prior knowledge assumed here was very high. Several pages of the manuscript are devoted to discussing Burgers Vectors, but Burgers Vectors were never defined/explained beyond "(slip plane)[Burgers Vector]" (line 215) , nor was slip system

terminology explained. I had to go to other sources to understand the concept, which, happily, were provided by the authors (e.g. Ref 32: Wheeler et al. 2009). I think more explanation would be beneficial. A few brief sentences would suffice, plus a more detailed explanation could be added to the supplementary material.

There are several labeling issues. The Inverse Pole Figure (IPF) plots are not labeled using a right handed system [i.e. using your right hand, 'a' (outstretched index finger) sweeps to 'b' (folded other fingers) when 'c' (thumb) is up]. The right-handed system should be strictly adhered to when working with crystals. (The Right-Handed System is being used by their Ref 49: Yung et al. 2006, if they want to see an example.) I suspect that MTEX has a changeable setting which will solve this problem.

The crystallographic directions in the olivine grains are never labelled. Axes labels would be useful for reference to slip planes and vectors. Crystallographic axes should be indicated in all Figures containing olivine grains (Figs. 2, 4, 6, 7). In fact, the convention for labelling olivine crystals was not given. Is it assumed to be Pbnm ($a = 4.76 \text{ \AA}$, $b = 10.20 \text{ \AA}$; $c = 5.98 \text{ \AA}$)? There are alternative choices in the literature (although less common) so this needs to be clarified.

Figure 7 is inadequately labelled. In Fig 7a, it would be useful to see Miller indices on the planes in the sketches, as well as indices for the low angle boundary and the Burgers Vector direction. The misorientation axis should also be labelled to help with our understanding. [The misorientation axis in the sketches appears to be [010] according to line 216 of the text, but subgrain slip boundaries visible on the EBSD images seem to be [100] according to line 283 of the text. The green N-S subgrain slip boundary in the pink box is called [100] in the Figure caption, but then the last sentence of the caption states that the misorientation axis is near [010]. This seems contradictory, but labels (or a clearer explanation of the labels) would solve this apparent problem.]

Furthermore, there are no crystallographic axes labelled on the olivine grain (in any figure, as stated previously) and the IPF is not a right handed system! In Fig 7d, X, Y and Z are used instead of a, b and c. This may simply be convention in Chemistry or Engineering, but a, b and c would be more consistent with olivine crystallographic axes. One final comment about Figure 7 is that it would be nice if the slip systems in 7e) were labelled A, B, C, D, E-type, beside the slip system symbols, for ease in making these connections with the text.

See additional questions, corrections, and clarifications in the accompanying annotated version of the manuscript.

Roberta L. Flemming

Reviewer #2 (Remarks to the Author):

An interesting paper that is sure to stimulate discussion in the petrology and volcanology communities. I am not an expert on the methods used here to look at dislocations but have considerable experience with Kilauea lava and its primary mineral, olivine. I have added comments on the word file for this paper, figures and a sample table. The tables need the most work for this manuscript.

The authors have done an excellent job researching the literature on Kilauea.

The interpretations presented here have fundamental implications for how magma is stored and deformation within volcanoes.

Review of NCOMMS-19-09434 by Wieser *et al.*

General comments

The manuscript addresses the origin of distorted olivine grains erupted from Kīlauea, for which a variety of disparate models have previously been proposed. This topic is important in deciphering the magmatic and petrological processes operating in the deep portions of the magmatic system, including the states, locations, movements, and evolution of partially solidified magma bodies. Distorted olivines are common features of mafic magmatic systems worldwide and therefore Kīlauea provides an important example of phenomena and processes that may be common. The authors take a new approach to characterising the olivines by employing electron backscatter diffraction (EBSD) to map their internal substructure, generating rich, quantitative datasets that go beyond those obtained previously by other techniques. The results point to a new model for the generation of distorted olivines involving deformation in crystal mush piles. The new model reconciles previously disparate petrological, geochemical, and geophysical observations and therefore the results will be of interest to workers in all these areas. As such, the general topic is suitable for *Nature Communications*.

The EBD work is extensive and contains novel aspects. One example is the automated incorporation of boundary traces into the classification of subgrain boundaries. The interpretation of the results is deep and rigorous. I have highlighted below several minor scientific points that would benefit from further clarification, but the overall scientific quality of the work is excellent. Therefore, I expect that the manuscript will be suitable for publication once the authors have had an opportunity to make minor revisions to address these points. I have also included several minor, less scientific, suggestions for the text, which are intended only to be helpful.

Sincerely,

David Wallis

Specific scientific comments

L40: Should this passage be “where olivine is the only silicate phase in magmas containing >6.8 wt% MgO”?

L76–79: You could be even more specific about the benefits of EBSD analysis. In particular, you can analyse large numbers of subgrain boundaries to improve statistical significance. This benefit is particularly potent in the determination of misorientation axes, which are imprecise when the misorientation angle is small (Prior, 1999; Wilkinson 2001) and therefore require large numbers of measurements.

L100: Replace “during textural re-equilibrium” with “during textural re-equilibration”.

L106: I’m not sure I understand the argument about truncation, which may be because I’m not sure that ‘truncation’ is the right word. The current text sounds as though some of the P-rich domain has been removed by the subgrain boundary, whereas it looks as though the subgrain boundary just happens to coincide with the divide between a P-rich core and P-poor rim. Perhaps some elaboration or rephrasing is necessary.

Figures 3a: Why the difference in the colour key between Figure 3a and those of figures 2 and 4a?

Figures 3c and 4c: State that the colour scales have units of ‘multiples of uniform distribution’.

L139: It would be better to cite original deformation experiments than Boioli’s modelling paper. How about Bai and Kohlstedt (1992, *Tectonophysics*)?

L193 onwards: The use of ‘fabric’ and ‘CPO’ could be clarified. You seem to be using “fabric” for the standard/idealised/endmember CPOs identified by Karato, Jung, *et al.* and “CPO” for other measured CPOs (e.g., “relationships between fabric types and crystallographic preferred orientations (CPOs)”). However, this is somewhat of a false dichotomy and further muddies the water regarding the (already ambiguous) term ‘fabric’. I would simply use CPO throughout and scrap fabric, which would be particularly beneficial in distinguishing when you are talking about CPO versus other microstructural elements indicative of particular slip systems, such as subgrain boundaries (which some workers would also include as ‘fabric elements’, but in a different sense to the use of ‘A-type fabric’ (meaning CPO) by Jung *et al.*).

L198: Lars’s paper on natural peridotites is not the best reference for a statement about deformation experiments. You already cite another of his papers, which does focus on Paterson experiments, so you could cite that here too instead.

L206: Lloyd *et al.* (1997, *Tectonophysics*) and de Kloe *et al.* (2002, *Microscopy and Microanalysis*) would be more appropriate references for determining slip systems from subgrain boundaries than Dave Prior’s EBSD review paper.

L209: “We propose that the proportions of different types of dislocation preserved in low angle boundaries reflect the relative activities of different slip systems.” This isn’t really your proposal. It’s an established idea that has been around for decades (e.g., references above), which is better for the strength of your argument anyway, so this sentence should be rephrased.

L214–215: Insert “edge dislocations”

L215: Replace with “twist walls consisting of [100] and [001] screw dislocations”

Figure 8a: I don’t think it makes sense to refer to subgrain boundaries as “A-type”, etc. Those terms were invented for CPO and using them here muddies the water again. I would wait until Figure 8c, in which you relate the slip systems to CPO types, instead.

L260: I don’t think you need to invoke the strain compatibility problem here. At relevant temperatures, olivine is essentially viscous, rather than plastic (meaning that the yield stress/CRSS is negligible). If you stress the crystal in an arbitrary orientation some strain rate will be produced on each slip system in proportion to their resolved shear stresses and viscosities.

L281: Change to “distance between subgrain boundaries”.

L297: Round the stress estimates.

L326: Replace “process” with ‘phenomenon’.

L328: Replace “parameter space” (which can’t generate mush piles) with ‘time scales’ or similar.

L337: Replace “InSAR displacements ... can be modelled as penny shaped cracks” with ‘can be modelled based on a magma chamber represented by a penny-shaped crack’.

L370: Replace “is thought to” with a statement of the evidence.

Figures 2, 3a, and 4a: State that the colour is based on the misorientation axis of each point relative to the mean orientation (if this is correct).

Text suggestions

L10 onwards: comma after “e.g.”

L33 onwards: Indicate a range of numbers with an en dash, rather than a hyphen.

L45: Update to “Materials with high seismic velocities”.

L47 onwards: Up to now you have used an Oxford comma to separate the final items in lists, but you have not done so after “events”. I recommend continuing the Oxford comma as it has no downsides and sometimes adds clarity.

L51: “south flank” should be hyphenated as a compound adjective. However, it would be better to avoid making a new jargon term, “south-flank instability”, and instead say “instability of the south flank”.

L63 and other instances: Avoid using “This” as an indefinite antecedent (not saying what ‘this’ is). Update to “This suite of eruptions...” or similar. There are many instances of similar indefinite antecedents throughout the manuscript.

L73: “oxidation-decoration” only needs to be hyphenated when used as a compound adjective (e.g., the oxidation-decoration technique), whereas here it does not.

L75 onwards: There’s no need to capitalise words to define an acronym.

L112: “were examined”... Up to now you have written in active voice but here you switch to passive voice. It’s better to remain in active voice as it’s more explicit (e.g., in this case who did the examining).

L124: Here, “low angle” is a compound adjective and therefore should be hyphenated.

L128: Replace “which” with ‘that’.

L167: “requires”

L198: comma after “that”

L222: “direction of the” is redundant.

L224: ‘within individual EBSD maps’

L236: Comma after “i.e.”

L275: Hyphenate the compound adjectives “subgrain-size and dislocation-density piezometry”.

L277 onwards: Scalar variables should be italic.

L358: Replace “deformation” with ‘deformed’.

L361: Replace “thought to” with ‘for which Pb isotopes indicate that they’

L367: Delete “glasses”.

L406: No need to capitalise “Electron Backscatter Diffraction”

L479: Replace “misorientated” with ‘misoriented’.

END OF REVIEW

**Reviewer 1- Roberta L. Flemming**

Wieser et al. have performed a careful study of strain in olivine in Kilauea eruptives (and
Greenland for comparison) and they have used that data together with several estimated
parameters to perform two piezometer calculations which allows them to interpret the
location of formation for their samples (using pressure, temperature and water content
estimates/assumptions), and furthermore to estimate the thickness of the mush pile. They
have done a considerable amount of work in this contribution, and the data appear
technically sound, and make an important contribution to the scientific community. The data
collection methods are not new, but the broad-ranging interpretation of their data advances
our understanding about the origins of deformed olivine. However, there are some
uncertainties in the interpretation, and the certain areas of the paper require clarification, as
outlined below.

The authors have done a good job of recognizing previous contributions to the literature. The
authors have done an excellent job of going back to the source for the techniques (e.g.
Donaldson 1976 olivine morphology; Toriumi 1979 piezometry; and Prior 1999 EBSD & OC
imaging). But what about the work by J.-C. J. Mercier, who wrote “Chapter 19. Olivine and
Pyroxenes” in the 1985 text book by Wenk, Ed. “Preferred Orientation in Deformed Metal
and Rocks. An introduction to Modern Texture Analysis” (1985), which shows textures, slip
systems, and pole figures for olivine in various settings, including piezometry.

**Unfortunately, we have been unable to get hold of this book, despite numerous**
**enquiries through the Cambridge university library system. We believe readers of this**
**paper will meet similar access issues if they try to find this book, so have favoured**
**the easily accessed references. We have also already exceeded the 70 reference limit**
**by adding other references suggested by the three reviewers.**

I think it would be useful to add optical photomicrographs where possible to Figures 2, 4, 6, 7, to compare with the EBSD images, to emphasize the improvement in quality of the data by EBSD.

Due to the highly vesicular nature of the scoria used in this study, thin section making was extremely difficult. Additionally, each thin section contains very few olivines (as vesicles occupy >90% of the area), so very large numbers of thin sections would have been needed to examine the large numbers of olivines in this study (329 grains, requiring ~100 thin sections). Thus, almost all EBSD maps were conducted on epoxy grain mounts (where ~50 olivines could be mounted at once). Only 1 thin section was successfully made from an epoxy stub (many attempts at this resulted in thin section fracture during the final polishing stages). Of the grains shown in Fig 4, only the bottom one was within this thin section. The other olivines that we show in Fig. 4 were chosen as they show the best examples of misorientation axes and several WBVD directions.

To address this reviewer comment, we have added a section in the supplementary information “Comparison of optical microscopy and EBSD” where we compare EBSD maps to optical photos from the grains present within the 1 available thin section. We have also added optical photos of the dendrites shown in Fig. 3

On Lines 76-77, the authors state that this work “offers significant improvements in spatial and angular resolution compared to optical observations”. On lines 606-607, they claim that using optical observations, misorientation boundaries < 1 degree (up to 2.5 degrees) were invisible. But the MTEX lower threshold of 0.5 degrees is arbitrary, and only one boundary needs to be identified (line 595).

We agree with the reviewer that the threshold angle used to define the presence of low angle boundaries is somewhat arbitrary. However, the 0.5° segmentation threshold chosen in this study represents what is conventionally seen as the smallest misorientation angles that can be resolved at reasonable precision given the finite angular resolution of conventional EBSD. Resolution estimates range from ~0.1-0.5°, with the higher resolutions available on modern EBSD detectors following the use of the Kuwahara filter (as used in this study; Humphreys, 2001, Brough et al., 2006-DOI: 10.1179/174328406X130902). Hansen and Warren (2015) also demonstrate that subgrain piezometry using 0.5° segmentation angles fall close to the experimental lines of Toriumi and Karato (Their Figure A1-d).

Conventional EBSD, like optical microscopy, will fail to identify lattice distortions with misorientations smaller than its angular resolution. The point of this section in the text was to emphasize that, even with this finite resolution, EBSD allows many low angle boundaries to be observed that would be missed by conventional optical examination of thin sections (see the images in the supplementary information).

The growing field of high angular resolution electron backscatter diffraction (HR-EBSD) enables misorientations of 0.006° degrees to be resolved (Britton and Hickey, 2018 - <https://iopscience.iop.org/article/10.1088/1757-899X/304/1/012003>), which would allow even more subtle distortions to be observed. However, this technique is difficult to apply to the large grain areas of the olivines examined in this study (David Wallis, personal communication), so at the moment, the 0.5° subgrain threshold chosen here is justified based on the resolution of applicable EBSD techniques. We are sure in future that further advances in EBSD will allow more subtle features to be identified within large grains. However, until subgrain-size piezometers are calibrated for the

159 **segmentation of subgrain boundaries at much smaller angles, it is wise to continue to**
160 **use established segmentation thresholds.**

**We have added more detail to the text to justify our chosen threshold angle (Lines**
**473-481):**

*“The classification of a grain as distorted or undistorted is somewhat arbitrary, particularly*
*based on optical observations^{25,26}. Here, the presence of distortion was quantitatively*
*assessed by identifying subgrain boundaries using the calcGrains MTEX function, with a*
*threshold angle of 0.5°. Olivines were classified as distorted if at least 1 low angle boundary*
*stretching across ~ a third of the crystal was identified. The use of smaller threshold angles*
*than 0.5° would result in imprecise estimates of misorientation axes and angles, due to the*
*finite angular resolution of conventional EBSD^{79,80}. Furthermore, utilization of smaller*
*threshold angles can result in inaccurate stress estimates from microstructural methods such*
*as subgrain piezometry⁵³”*

I would like to see more clarity in the pressure obtained by their observations. Their
piezometry calculation suggests relatively low pressure (~3-12 MPa), allowing them to
discount rift zone origin,

**We thank the reviewer for pointing out that the reason we have discarded a deep rift**
**zone origin was unclear in the original text. We suggest that it is unlikely that olivines**
**are derived from the deep rift zones for three main reasons. The strongest line of**
**evidence for their origin from mush piles within the South Caldera Reservoir comes**
**from the spatial distribution of eruptions containing large numbers of deformed**
**grains. This is outlined in the conclusion. The second line of evidence against a deep**
**rift zone origin is the fact that olivines don't show adcumulate textures. Thirdly,**
**deformed olivines are seen at volcanoes that lack deep rift zones (e.g. Reunion,**
**Iceland), but likely contain mush piles at the base of magma reservoirs.**

**We have addressed these points in the text between lines 193-224:**

*“To generate lattice distortions, crystals must be subject to non-hydrostatic stress⁶. Previously,*
*olivine deformation has been attributed to differential stresses exerted within Kīlauea's deep*
*rift zone dunites, during the downward flow of olivine from the summit area to the rift zones,*
*or rapid rift zone extension during south flank earthquakes¹⁵. However, mechanisms involving*
*deep rift zone processes require specific magma transport paths to bring deformed crystals to*
*the surface. For example, the presence of deformed grains in lavas from the Mauna Ulu*
*eruption have led to suggestions that magma is transported along the basal decollement⁸.*
*However, significant magma transport through the deep rift zones is at odds with geophysical*
*evidence, which demonstrates no measurable changes in rift zone opening rates during*
*magma supply surges or rift zone eruptions¹⁸ and generally supports shallow (< 2 km) magma*
*transport⁴⁵. Given the wide spatial and temporal distribution of deformed grains identified in*
*this study, it seems untenable that every eruption received a significant contribution to its*
*olivine crystal cargo from deep rift zone transport paths.*

*The textural features of deformed olivines are also inconsistent with an origin from low melt*
*fraction dunites (<5 % melt)¹⁵. Olivines derived from such bodies would be expected to show*
*adcumulate textures similar to those found in xenoliths erupted at Kīlauea Iki⁶ and post-shield*
*Hawaiian lavas⁴⁶. Instead, most distorted grains have subhedral-euhedral outlines with no*
*internal melt films and high angle grain boundaries (Fig. 4). While corrosion followed by*
*overgrowth can produce euhedral crystals from disaggregated dunites⁴⁷, this process would*
*preserve internal melt films.”*

*“A crucial observation is the fact that deformed olivines have been noted in a variety of*
*basaltic systems. Vinet et al.²⁵ report their presence in all five of the volcanic centres which*
*they examined in the Andean Southern Volcanic Zone. Deformed olivines have also been*
*noted in the Vaigat formation of West Greenland²², the Baffin Bay volcanic province⁴⁸, Piton*
*de la Fournaise volcano⁵, and the Loch Uisg area, Mull⁴⁹. This implies that olivines are*
*deformed by a process which is near-ubiquitous in a wide range of volcanic plumbing*
*systems. We suggest that a plausible mechanism is gravitational loading within melt-rich*
*cumulate piles with ~40% porosity⁵⁰ produced by crystal settling². Such an origin accounts*
*for the global distribution of deformed grains, and the absence of adcumulate textures at*
*Kīlauea. To investigate this hypothesis further, we assess the location, and conditions under*
*which deformation occurred.”*

**We also clarify these points in the conclusion (Lines 432-440):**

*“Previous petrological work has concluded that differential stresses are applied during the*
*creep of dunitic bodies within Kīlauea’s rift zones. Crucially, this has fuelled suggestions that*
*significant quantities of magma bypass the summit reservoir and travel along the deep rift*
*zones, despite the paucity of geophysical evidence for this process. In contrast, we suggest*
*that olivine crystals are deformed in gravitationally-loaded olivine mush piles at the base of*
*the deeper South Caldera reservoir, explaining the spatial distribution of grains at Kīlauea, and*
*the absence of adcumulate textures. Moreover, the mechanism we propose here accounts for*
*the occurrence of deformed grains in a wide variety of basaltic systems worldwide, which lack*
*deep rift zones.*

**We then utilize subgrain piezometry to estimate whether it is possible for**
**gravitationally loading within mush piles to generate the observed microstructures**
**(and using the simple box model in figure 10, these stresses can indeed be generated**
**utilizing known parameters regarding storage geometry and magma supply rates at**
**Kīlauea; summarized in the conclusion in lines 442-446):**

*“Application of piezometers developed for mantle peridotites reveals that deformed olivines*
*experienced differential stresses of ~3–12 MPa. Assuming that olivine mush piles behave as*
*granular materials, and form force chains, the vertical extent of mush piles at Kīlauea can be*
*estimated (~180-720 m). Based on available constraints on the magma supply rate, and the*
*geometry of Kīlauea’s summit reservoir, these thicknesses accumulate in a few centuries.”*

but there is uncertainty over interpretation of one set of slip directions as either {0kl}[100] slip
(high stress, low T, low H₂O – line 266) or a combination of (010)[100] and (001)[100] - line
268 (indicating deformation at mod-high T, low-mod H₂O – lines 272-273) . These two sets
of observations seem to lead to a circular argument:

Lines 302-308 – “The apparent presence of {0kl}[100] slip is at odds with our differential
stress and water content estimates (Fig. 8c), supporting our earlier suggestion that subgrain
walls apparently relating to {0kl}[100] slip are in fact made from mixtures of (010)[100] and
(001)[100] type dislocations. Our estimates for water contents and differential stress, which
indicate conditions where the strengths of (010)[100] and (001)[100] slip are comparable, are
then entirely consistent with our microstructural analysis of slip system proportions.”

If the active slip system was assumed to be {0kl}[100], would this provide a higher calculated
pressure using dislocation piezometry? It would be useful to be able to definitively test the
existence of {0kl}[100] slip in order to clarify their argument.

**We thank the reviewer for pointing out that this argument was unclear in the original**
**manuscript. We have now emphasized that the conditions of deformation estimated**
**from analysis of subgrain wall geometry (e.g. misorientation axis, WBVD, boundary**
**strike) are independent of the piezometry performed on linear intercept distances and**

dislocation densities. The stress estimates obtained from linear intercept and
dislocation density piezometry are too low for $\{0kl\}[100]$ slip to be favoured. Thus, as
suggested for mantle peridotites, it is highly probable that the apparent presence of
$\{0kl\}[100]$ slip is due to the mixing of two different edge dislocations along a single
boundary. Sadly, there is no simple way to resolve $\{0kl\}[100]$ dislocations from
combinations of $(010)[100]$ and $(001)[100]$ slip using conventional EBSD.

**We have addressed this in the text in lines 258-266:**

*“The conditions producing D-type fabrics are debated, they may involve high stresses, low*
*temperatures and low water contents favouring slip on $\{0kl\}[100]$ ^{29,32}. Alternatively,*
*crystallographic signatures indicative of $\{0kl\}[100]$ slip may be produced when $(010)[100]$*
*and $(001)[100]$ edge dislocations are present within a single subgrain boundary³⁶. Overall,*
*the dominance of slip systems with $[100]$ WBVD in Kīlauean olivines is indicative of*
*deformation at moderate-high temperatures and low-moderate water contents (Fig. 8a). To*
*further constrain the deformation conditions, and assess the origin of $\{0kl\}[100]$ slip, we*
*place independent constraints on differential stresses and olivine water contents.”*

**Lines 304-316:**

*“The low differential stresses estimated from linear intercept distances and dislocation*
*densities are difficult to reconcile with the conventional interpretation that misorientation axes*
*lying between $[001]$ and $[010]$ represent slip on the relatively high stress $\{0kl\}[100]$ system.*
*Instead, our data support the hypothesis that a $\{0kl\}[100]$ -like misorientation axis distribution*
*is generated by the presence of $(010)[100]$ and $(001)[100]$ edge dislocations within a single*
*subgrain boundary³⁶. Slip on $(010)[001]$ dominates at high stress and/or moderate-high water*
*contents, while $(100)[001]$ slip dominates at high water contents and low stress⁵⁶ (Fig. 8c).*
*Our independent estimates of olivine water contents and differential stresses lie at this*
*transition, where the strengths of $(010)[100]$ and $(001)[100]$ slip are comparable³⁰. Thus, it is*
*highly probable that subgrain walls in Kīlauean olivines apparently relating to $\{0kl\}[100]$ slip*
*are actually made from mixtures of $(010)[100]$ and $(001)[100]$ type dislocations. This reconciles*
*our independent estimates of deformation conditions from subgrain piezometry and the*
*microstructural analysis of slip system proportions.”*

The force chain argument also seems like a stretch here. The problem with invoking this
theory here is that there is no evidence in the data measured to test support for this
proposed mechanism. Only single grains are observed (not a dunitic texture) so the
proposed force chain mechanism can not be confirmed by observations within this context.

**We appreciate that as we are not observing force chains in-situ, it is hard to provide**
**direct evidence that they form within the base of the South Caldera Reservoir. It is**
**worth clarifying that we suggest that force chains form within relatively melt-rich**
**cumulate piles (~40% porosity) produced by the settling of olivine crystals from**
**evolving melts due to their high density. The high melt fractions in such cumulate**
**piles would not result in the development of a dunitic texture (which would likely only**
**form in melt-poor cumulate piles). Lines 220-223:**

*“We suggest that a plausible mechanism is gravitational loading within melt-rich cumulate*
*piles with ~40% porosity⁵⁰ produced by crystal settling². Such an origin accounts for the*
*global distribution of deformed grains, and the absence of adcumulate textures at Kīlauea.”*

**However, the unequal transmission of force has been recognised as a near ubiquitous**
**property of granular materials. The development of force chains has widespread**
**implications for fruit vendors, the storage of grain within corn silos, and many other**

industrial applications (see Bergantz et al., 2017; for a summary). This well known
phenomenon was first quantified using photo-electric disks, and later with DEM
models (as used in Bergantz et al., 2017). Thus, although we cannot prove that force
chains form within granular olivine mush piles, we suggest that is highly probable
that piles of olivine grains act like many other granular materials (rather than having
some unique property that prevents the formation of force chains).

**We have added this argument into the main text in lines 329-336:**

*“However, centuries of observations have demonstrated that a near-ubiquitous feature of*
*granular materials is the arrangement of solid particles into force chains, resulting in unequal*
*transmission of load (e.g. corn in silos, stacked fruit, poppy seeds; photoelastic particles)⁵⁹⁻*
*62. Highly stressed chains of primary members transmit most of the force, while a larger*
*group of secondary members experience less than the mean force, and a few spectator*
*particles experience no force (Fig. 9b)⁶³. The settling of dense olivine crystals into mush*
*piles creates a crystal framework with ~40% porosity⁵⁰, which constitutes a granular*
*medium. Thus, it is highly probable that mush piles contain force chains.”*

**And in the conclusion (Lines 443-445):**

*“Assuming that olivine mush piles behave as granular materials, and form force chains, the*
*vertical extent of mush piles at Kilauea can be estimated (~180-720 m).”*

I like the authors' suggestion (Lines 354-356) “An alternative research direction would be to
perform EBSD on olivines at the base of mush piles of known thickness (e.g. exposed sills)
to produce a calibration between deformation intensity and mush pile thickness.” but
apparently, observations of the base of mush piles are uncommon (line 401).

**Unfortunately, the reviewer is correct; before erosion of Hawaiian islands reach the**
**depths where mush piles are likely to be present (3-5 km below the shield stage**
**lavas), the volcanic edifices subside below sea level (Tom Shea, Pers Comms, 2019).**
**We have now clarified that we were suggesting the examination of mush piles in**
**much older, extinct volcanic systems where erosion/mass wasting/faulting has**
**caused solidified mush piles to outcrop at the surface (Lines 353-356).**

*“An alternative research direction would be to perform EBSD on olivines at the base of mush*
*piles of known thickness in extinct volcanic systems where intrusions outcrop at the surface*
*to produce a calibration between deformation intensity and mush pile thickness.”*

Clarity of the writing is also an issue. I realize that Nature Communications caters to experts
in the field, but the level of prior knowledge assumed here was very high. Several pages of
the manuscript are devoted to discussing Burgers Vectors, but Burgers Vectors were never
defined/explained beyond “(slip plane)[Burgers Vector]” (line 215) , nor was slip system
terminology explained. I had to go to other sources to understand the concept, which,
happily, were provided by the authors (e.g. Ref 32: Wheeler et al. 2009). I think more
explanation would be beneficial. A few brief sentences would suffice, plus a more detailed
explanation could be added to the supplementary material.

**We thank the reviewer for highlighting the lack of introductory material regarding slip**
**systems. We have added a new section to the introduction where we lay out the basic**
**information regarding dislocation motion, and its relation to slip systems. We also**
**clarify the connection between our study and extensive experimental work by the**
**mantle peridotite community (Lines 87-106).**

*“If distorted olivines record plastic deformation, rather than crystal growth processes, their*
*microstructures can be evaluated within the conceptual framework developed through the*
*extensive study of naturally- and experimentally-deformed mantle peridotites. These studies*
*have identified five olivine fabric types (A to E-types) with different crystallographically*
*preferred orientations (CPOs), and determined the role that pressure, temperature, differential*
*stress and hydration exert on their formation*^{29–31}. *However, unlike thin sections of mantle*
*peridotites, the orientations of erupted olivines in tephra and spatter deposits have been*
*randomised upon eruption and again during sample preparation, so CPOs cannot be used to*
*infer deformation conditions in volcanic olivines. However, CPOs are macroscopic features*
*resulting from the activities of different dislocation types (which are classified in terms of slip*
*systems)*^{30,32–35}.

*The movement of dislocations is one of the main mechanisms accommodating plastic*
*deformation within crystals. Dislocations can be characterized in terms of a Burgers vector (\vec{b})*
*and a line direction (\vec{u}). The plane containing these vectors defines the slip plane³⁶. The angle*
*between \vec{b} and \vec{u} determines whether the dislocation is an edge dislocation ($\vec{b} \perp \vec{u}$), a screw*
*dislocation ($\vec{b} \parallel \vec{u}$), or a mixed dislocation (\vec{b} and \vec{u} neither parallel or perpendicular). A slip*
*system is described in terms of a set of slip planes, and a Burgers vector direction on which*
*dislocation motion occurs. For example, the olivine slip system (010)[100] denotes the*
*movement of edge dislocations along the (010) plane with a [100] Burgers vector direction. “*

**We have also restructured the manuscript to move the detail regarding the new code**
**developments into the methods section, so that the main text is more accessible to**
**readers with limited EBSD experience.**

EBSD maps of individual olivines provides sufficient crystallographic information to quantify
the Burgers vector and slip plane of low angle boundaries (see methods). As many
deformation experiments were conducted at conditions similar to those found within igneous
plumbing systems (e.g., 250–300 MPa, 1200°C)³⁷, assessment of the activities of different
slip systems places constraints on the conditions of deformation (using established links
between slip systems, CPOs and fabric types, and between fabric types and deformation
conditions)^{30,32–35}.

.
There are several labeling issues. The Inverse Pole Figure (IPF) plots are not labeled using
a right handed system [i.e. using your right hand, ‘a’ (outstretched index finger) sweeps to ‘b’
(folded other fingers) when ‘c’ (thumb) is up]. The right-handed system should be strictly
adhered to when working with crystals. (The Right-Handed System is being used by their
Ref 49: Yung et al. 2006, if they want to see an example.) I suspect that MTEX has a
changeable setting which will solve this problem.

**Following communication with the developers of MTEX, it seems that both right and**
**left handed co-ordinate systems are used on publications of olivine deformation (as**
**these are equivalent in orthorhombic systems). However, to follow the reviewers**
**advise, we have altered our figures to reflect the RH convention.**

The crystallographic directions in the olivine grains are never labelled. Axes labels would be
useful for reference to slip planes and vectors. Crystallographic axes should be indicated in
all Figures containing olivine grains (Figs. 2, 4, 6, 7).

**We have added 3D models of olivine grains (using the crystalShape MTEX package)**
**onto each figure. We have chosen to use these instead of crystallographic axes as we**
**believe it is easier to visualize the 3D nature of olivine orientation on a 2D image using**
**these (see example below). They also stand out more than arrows on the images of**

deformed olivines where lots of colors are used to show crystallographic orientations.
 In each instance for deformed grains, we have shown a single olivine illustration for
 the mean orientation of the crystal (the small misorientations do not produce visible
 changes in the orientations of arrow or 3D olivines). For the dendritic olivines shown
 in Fig. 3, we have shown 1 3D crystal for the mean orientation of the grain highlighted
 with the red dot.

In fact, the convention for labelling olivine crystals was not given. Is it assumed to be Pbnm
 ($a = 4.76 \text{ \AA}$, $b = 10.20 \text{ \AA}$; $c = 5.98 \text{ \AA}$)? There are alternative choices in the literature
 (although less common) so this needs to be clarified.

**We have used Pbnm, we have clarified this in the caption for figure 2 (Lines 574-576):**

*“Light blue 3D olivine crystals are superimposed (using the MTEX crystalShape package)
 to allow visualization of the orientation of the three crystallographic axes ([100], [010], and [001],
 space group Pbnm).”*

Figure 7 is inadequately labelled. In Fig 7a, it would be useful to see Miller indices on the
 planes in the sketches, as well as indices for the low angle boundary and the Burgers Vector
 direction. The misorientation axis should also be labelled to help with our understanding.

[The misorientation axis in the sketches appears to be [010] according to line 216 of the text,
 but subgrain slip boundaries visible on the EBSD images seem to be [100] according to line
 283 of the text. The green N-S subgrain slip boundary in the pink box is called [100] in the
 Figure caption, but then the last sentence of the caption states that the misorientation axis is
 near [010]. This seems contradictory, but labels (or a clearer explanation of the labels) would
 solve this apparent problem.] Furthermore, there are no crystallographic axes labelled on the
 olivine grain (in any figure, as stated previously) and the IPF is not a right handed system! In
 Fig 7d, X, Y and Z are used instead of a, b and c.

**We think these comment results from our lack of clarification in the text that the
 WBVD for subgrain boundaries, and the misorientation axis, are both denoted as
 crystallographic directions (e.g. [100]). We have also now drawn both co-ordinate
 systems onto Fig. 7c to emphasize the differences, and discussed the differences
 between these co-ordinate systems in the figure caption (Lines 651-653):**

*“Specimen coordinates (orange arrows) represent the x, y, and z directions within the SEM.
 Crystal coordinates (pink arrows) represent the directions of the [100], [010], and [001] axes
 of the olivine.”*

“In Fig 7a, it would be useful to see Miller indices on the planes in the sketches, as well as
indices for the low angle boundary and the Burgers Vector direction. The misorientation axis
should also be labelled to help with our understanding.”

**We have also attempted to clarify in the figure caption that figure 7a and b show**
**schematic illustrations of a tilt and twist boundary, and are not related to the**
**deformed grain shown in figure c (by adding “schematic” to the titles of these part**
**figures; Lines 646-650).**

*“a) A schematic diagram of a tilt boundary, where the rotation axis lies within the plane of*
*the low angle boundary. b) A schematic diagram of a twist boundary, where the boundary*
*plane is perpendicular to the rotation axis. Two sets of screw dislocations are shown within*
*the twist wall for illustrative purposes; this is just one possible geometry.”*

Fig 7d, X,Y and Z are used instead of a, b and c. This may simply be convention in Chemistry
or Engineering, but a, b and c would be more consistent with olivine crystallographic axes.

**In this figure, we have shown specimen co-ordinates (X,Y,Z of the SEM stage) as well**
**as crystallographic co-ordinates ([100], [010], [001]. We have tried to clarify this by**
**adding more annotations to Figure 7c and more detail to the figure caption (Lines 646-**
**665):**

“Fig. 7 –Slip system identification method developed for this study. a) A schematic diagram of a tilt boundary, where the rotation axis lies within the plane of the low angle boundary. b) A schematic diagram of a twist boundary, where the boundary plane is perpendicular to the rotation axis. Two sets of screw dislocations are shown within the twist wall for illustrative purposes; this is just one possible geometry. c) Map of WBVDs in a deformed olivine. Specimen coordinates (orange arrows) represent the x, y, and z directions within the SEM. Crystal coordinates (pink arrows) represent the directions of the [100], [010], and [001] axes of the olivine. A 3D olivine crystal is overlain to demonstrate the orientation of the crystal in crystal co-ordinates. Boundaries with [100] WBVDs are almost parallel to the y direction, while those with

[001] WBVDs are almost parallel to the x direction in specimen coordinates. A small section
of a [100] WBVD boundary is selected for further discussion (pink rectangle). d) Ideal tilt
(yellow) and twist (red) boundaries calculated for each boundary segment in the pink rectangle
based on the trace of the subgrain boundary and direction of the misorientation axis (in
specimen coordinates). As the calculated twist wall planes show extremely low dips, the MTEX
code classifies all segments along this boundary as tilt walls. e) Schematic diagram showing
the misorientation axes in crystal coordinates for tilt boundaries built from dislocations types
associated with each of the 5 olivine slip systems (written as (slip plane)[Burgers vector]),
allowing 20° leniency. Misorientation axes of the boundary pixels within the pink rectangle plot
near [010]. Along with the [100] WBVD, this indicates that this tilt wall consists of (001)[100]
dislocations. “

One final comment about Figure 7 is that it would be nice if the slip systems in 7e) were labelled A, B, C, D, E-type, beside the slip system symbols, for ease in making these connections with the text.

We appreciate that it is slightly confusing that we refer to the slip systems in terms of their (slip plane) [Burgers vector] until we discuss figure 8c. However, this reflects the facts that the terms A to E-type fabrics were developed for CPOs, and so are not really applicable to our method where slip systems are identified from the crystallographic properties of sub-grain boundaries. Linking our observed fabrics to known CPO fabrics is only useful when evaluating the deformation conditions. This dilemma in naming conventions is summarized in the comments from reviewer 3, who state: *Figure 8a: I don't think it makes sense to refer to subgrain boundaries as "A-type", etc. Those terms were invented for CPO and using them here muddies the water again.*

See additional questions, corrections, and clarifications in the accompanying annotated version of the manuscript (see screenshots below).

Olivine water content ($H/10^{25}Si_{\text{atomic}}$)

Fig. 8 –Quantifying the conditions of deformation. **a)** Proportions calculated by summing the total length of each boundary type in 62 deformed grains (LHS), and by averaging the proportion of each slip system per grain (RHS; giving equal weight to all grains regardless of their size). **b)** Differential stresses calculated using subgrain piezometry⁵⁰ applied to linear intercept distances clearly overlap with those estimated from dislocation density piezometry⁵¹ on published dislocation densities²⁵. **c)** Olivine regime diagram⁴⁹ for T=1470-1570K with subgrain stress estimates (purple bar) and olivine water contents (orange bar) superimposed. **[What is your reference for the primary vs secondary slip? Poirier 1975?]**

We thank the reviewer for highlighting that it was unclear that we have inferred that [100] slip is primary based on the occurrence of this WBVD in all olivines, while [001] slip is only present in grains also containing [100] WBVDs (suggesting that this is a secondary slip system). We have clarified this in lines 247-251.

"Boundaries related to (010)[100], {0kl}[100] and (001)[100] slip comprised ~30%, ~44% and ~16% of the total length of tilt boundaries respectively (Fig. 8a). This implies that [100] slip systems represent the primary slip systems, with the mosaicity of crystallographic orientation in some olivine grains (Fig. 4) resulting from secondary (100)[001] and (010)[001] slip (~3 and ~6% of boundaries respectively; Fig. 8a)."

We have also added this detail into the figure caption for Fig. 8 (Lines 670-672):

"All grains contain [100] WBVDs, suggesting that this is the primary slip direction. Only a subset of grains contain low angle boundaries with [001] WBVDs, suggesting that this is a secondary slip direction."

Fig. 10 –Estimating mush pile accumulation rates using a simple model. **a)** A cylinder is supplied with primary melt (16.5 wt% MgO) at a rate of 0.1 km³/yr^[59]. Fractionation of this melt to the output composition (taken as the average whole rock MgO content of erupted lavas; ~10 wt% MgO) requires fractionation of 14 vol % olivine crystals¹⁵. **b)** The interplay between cylinder radius and accumulation time is shown for mush piles with thicknesses of 200 m and 700 m (assuming 40% porosity). Available constraints on reservoir radius from InSAR are overlain⁶⁰. **[Spell out InSAR]**

Amended

684 27. Humphreys, F. J. Review Grain and subgrain characterisation by electron backscatter diffraction.

685 *Journal of materials science* **36**, 3833–3854 **[date?]**

We thank the reviewer for noticing this omission, we have now added the date (2001).

Reviewer 2 –Anonymous

Reviewer #2 (Remarks to the Author):

An interesting paper that is sure to stimulate discussion in the petrology and volcanology
communities. I am not an expert on the methods used here to look at dislocations but have
considerable experience with Kilauea lava and its primary mineral, olivine. I have added
comments on the word file for this paper, figures and a sample table. The tables need the
most work for this manuscript.

The authors have done an excellent job researching the literature on Kilauea.

The interpretations presented here have fundamental implications for how magma is stored
and deformation within volcanoes.

We thank the reviewer for their overall support of this manuscript and its novelty.

**Comments on manuscript (screenshotted below from pdf comparison software –
585 RHS, submitted version, LHS reviewer comments).**

may provide an unexplored source of information. However, the magmatic processes
producing distorted olivines remain poorly understood. Previously, these features have been
described as subgrains and kink bands; we **initially** favour the descriptive term "distorted" to
avoid any genetic connotations.

may provide an unexplored source of information. However, the magmatic processes
producing distorted olivines remain poorly understood. Previously, these features have been
described as subgrains and kink bands; we favour the descriptive term "distorted" to **avoid**
any genetic connotations.

We have removed the term initially from this sentence

126 microstructure.
127 Most dendrite buds show identical crystallographic orientations to the primary crystal;
128 merging of such buds cannot produce lattice distortions. The 125 **crystal buds which are**
129 **misonorientated with respect to their host crystal (Fig. 3a)** show a bimodal distribution of

127 Most **(7%) dendrite buds in the Greenland (7) samples analyzed** show identical
128 crystallographic orientations to the primary crystal; merging of such buds cannot produce
129 lattice distortions. The 125 **misonorientated crystal buds** show a bimodal distribution of
130 misonorientation axes, with a strong maximum around [010]³⁸ and a weaker maximum around

**We have clarified this sentence to make it clear which samples the dendritic
measurements were taken in (Lines 151-152).**

*"Most branching dendritic buds in the West Greenland olivines show identical
crystallographic orientations to the primary crystal"*

investigate the location of gravitationally loaded mush piles. Geophysical and geochemical observations at Kilauea have led to a model in which magma is stored at two distinct depths⁵⁵⁻⁵⁸. We find the largest proportion of deformed olivines in eruptions thought to derive a significant proportion of their magma supply from the deeper (South Caldera) reservoir based on Pb isotopes^{54,55}. In contrast, there are few deformed grains in intracaldera and

investigate the location of gravitationally loaded mush piles. Geophysical and geochemical observations at Kilauea have led to a model in which magma is stored at two distinct depths⁵⁵⁻⁵⁸. We find the largest proportion of deformed olivines from eruptions thought to derived from the deeper (South Caldera) reservoir based on Pb isotopes^{54,55}. In contrast, there are few deformed grains in intracaldera and volcanic SWRZ eruptions thought to tap the upper, more evolved (Halema'uma'u) reservoir. The different source of these eruptions is

We have replaced this sentence with “ removed the term initially from this sentence (Lines 360-361):

“We find the largest proportion of deformed olivines in eruptions derived from the deeper South Caldera (SC) reservoir based on Pb isotopes (Pietruszka et al., 2018, 2015).”

is a simpler explanation for their abundance in a wide variety of subaerial eruptions than invoking magma transport paths along the basal decollement⁶. The narrow range of forsterite contents in deformed grains (~Fo₇₅; Fig. 5) may result from diffusive re-equilibration of Mg and Fe during prolonged mush pile storage⁸. The scavenging of deformed grains from mush

is a simpler explanation for their abundance in a wide variety of subaerial eruptions than invoking magma transport paths along the basal decollement⁶. The narrow range of forsterite contents of most deformed grains in our study (~Fo₇₅; Fig. 5) may result from diffusive re-equilibration of Mg and Fe during prolonged mush pile storage⁸. The scavenging of deformed grains from mush piles just before eruption also explains the high degree of olivine-liquid

We have added “in our study” to this sentence.

study (~3-12 MPa). Deformed olivines from the ~1700 AD eruption may record the first stages of this process; they show prominent subgrains, and high angle grain boundaries (misorientations >10°; Fig 6b-c). The latter are decorated with thin films of melt (Fig. 6b), indicating derivation from low melt fraction bodies. These olivines also show trails of secondary vapor bubbles, similar to those observed in Hawaiian xenoliths¹⁵. Finally, these

study (~3-12 MPa). Deformed olivines from the ~1700 AD eruption may record the first stages of this process; they show prominent subgrains, and relatively high angle grain boundaries (misorientations >10°; Fig 6b-c). The latter are decorated with thin films of melt (Fig. 6b), indicating derivation from low melt fraction bodies. These olivines also show trails of secondary vapor bubbles,

We have added the suggested paragraph break. We have not added the term “relatively” as in EBSD studies the terms high and low grain boundaries have specific definitions (now defined in lines 415-417).

“Deformed olivines from the ~1700 AD eruption may record the first stages of this process; they contain prominent low and high angle boundaries (misorientations <10° and >10° respectively; Fig 6b-c).”

Comments on tables (screenshots below)

✗ Poorly formatted tables in supplement

what is this sample

Inconsistent and Excessive Significant

Deformed/Undeformed	Grain Size	Sample	Fo	SiO2	MgO	CaO	FeO
Deformed	Big	908	0.851893	39.8892	45.1018	0.22	13.9773
Deformed	Big	908	0.851878	39.8791	44.9044	0.219077	13.9177
Deformed	Big	908	0.875526	40.3076	46.5807	0.223846	11.8047
Deformed	Big	908	0.876356	40.5096	46.9505	0.197615	11.8079
Deformed	Big	908	0.876193	40.1788	47.0083	0.203769	11.8402
Deformed	Big	908	0.875715	40.5089	46.1615	0.196154	11.6781
Deformed	Big	908	0.875718	40.5089	46.9749	0.209538	11.8836
Deformed	Big	908	0.878756	40.5873	47.0332	0.198077	11.5674
Deformed	Big	908	0.875823	40.2788	46.9141	0.194231	11.8568
Deformed	Big	908	0.875445	40.3294	46.8057	0.212077	11.8705

We have reformatted the tables to show 2 dp (3 for Fo content). We have added the “KL0” to the sample #, and added a line to the top of each sheet stating that further sample details are available in the first sheet.

Big
Big
Big
Big
Big
Big
Big
Big
Big
Big
Small
Small
Small
Small
Small
Small
Small
Small
Small

almost useless size info

We have replaced "big" and "small" with > and < 1mm.

Comments on figures (screenshots below)

441 Figure 1

not readable
Total # of dislocations needed
what is "n" for each location

basemap too dark

We have changed the opacity of the baselayer to make it less dark. We have also
added the # of deformed and undeformed grains from each eruption so that the
number of grains examined for each eruption can be easily evaluated (revised figure
below).

Fig. 1 – Spatial distribution of distorted grains across the Kīlauean edifice. Bar charts show the proportion of distorted grains across all sizes, and subdivided into grains >1mm and <1mm (based on sieved size fractions). Basemap from GEOMAPP APP.

15 = $\frac{137 \text{ dislocations}}{9 \text{ rocks/eruptions}}$
 ↑
 small # for each eruption

~~Distorted grains in samples~~

We feel it is important to clarify that what we have shown on the y axis of each figure
 is the % of distorted grains in each eruption. The number of dislocations is far higher,
 as most deformed grains contain at least 3 low angle boundaries that run across the
 entire length/width of the grain. While the number of deformed grains from each
 eruption is relatively low, we have collected EBSD maps on 329 olivines (of which 126
 were deformed, 203 were not deformed). This is the most comprehensive database
 collected on any volcano to our knowledge. Additionally, this represents more than 82
 648 hours of manual data collection on the SEM (plus significant additional instrument
 time to repeatedly calibrate the EBSD detector). While we agree that the absolute %'s
 of deformed grains in each eruption may change somewhat if more acquisitions were
 collected, we feel the number of grains analysed for each eruption is high enough to
 be confident that intracaldera eruptions contain far fewer deformed grains than
 extracaldera/rift eruptions (which is the main point that this figure is meant to
 convey). These numbers are mostly meant as a guide.

462 Figure 2

The aim of the figure was to demonstrate that deformed grains have poor to no P zoning (as emphasized by the reviewer in their comments). We have clarified in the figure caption and text that P zoning is absent in these crystals, which is inconsistent with the dendritic growth theory (Lines 137-139):

“However, wavelength-dispersive spectroscopy (WDS) maps collected by electron microprobe reveal homogenous cores within almost all distorted grains, (Fig. 2).”

(Lines 564-568):

“a-c) Grains with internal lattice distortions (revealed by stripes of different colors in EBSD maps) show homogenous concentrations of P in their cores. Observed phosphorus enrichments in the rim of (c) likely marking a period of rapid growth upon eruption (c). In (d), internal P enrichments are truncated against a subgrain boundary.”

In response to several reviewer comments, we have emphasized that the cause of P truncation in d) is uncertain; the point of including this was to counter the statement of Welsch et al. that P zones are not truncated against low-angle boundaries (Lines 140-145).

“One P-rich zone terminates against a subgrain boundary (Fig. 2d), suggesting that P
 enrichments and subgrains did not form during the common process of dendritic growth.
 Instead, the rapid growth episode generating the P zoning may have preceded the distortion
 of the crystal lattice, perhaps separated by a dissolution episode (c.f., ref. ⁵). Thus, while some
 olivines at Kīlauea may have experienced an early dendritic growth phase, this process
 appears unrelated to the production of lattice distortions.”

with crystallographic directions deviating from the mean are colored. ^{For} ~~Once the~~ deviation
 angle ^{is} greater than $>3^\circ$, the pixel is colored black. Most distorted grains show no P
 enrichments in their cores (a, b, c). Observed phosphorus enrichments occur in the rims of
 some grains, likely marking a period of rapid growth upon eruption (c). In some cases,
 internal P enrichments are clearly truncated by subgrain boundaries (d).

We have amended this figure caption to read (lines 573-574):
 “For deviation angles greater than $>3^\circ$, the pixel is colored black”

**Figure 3**
 **Fig. 3 – Crystallographic signatures of dendritic branching in West Greenland picrites²². a)**
 **EBSD maps colored using the IPF key, with the mean orientation of the grain containing the**
 **red dot set to white. Pixels with deviation angles $>20^\circ$ are colored black. b) Weighted Burgers**
 **Vector directions (WBVD) overlain on band contrast maps ([100]=green, [010]=blue,**
 **[001]=red)). c) Misorientation axes for 125 misorientated buds (misorientation angle $>1^\circ$). d)**
 **Histogram of misorientation angles between adjacent dendrite buds. Many buds are**
 **misorientated from the host crystal by $>>5^\circ$.**

*I label plots as in Fig. 4

 We thank the reviewer for the comments improving this figure. We have increased the
 font size on this figure, and added labels as in Figure 4. We have also clarified that the
 misorientated dendrites are from Greenland, and added that 70% of grains are
 misorientated by $>5^\circ$, and defined EBSD and IPF in the caption.

The poor image quality in b is a result of the fact that this is a map of band contrast,
 and its “blurriness” results from the pixel size at which acquisitions were collected
 (the same reason the EBSD maps in Fig. 2 (RHS) are pixelated).

486

Figure 4

Neat method

**Fig. 4** – Crystallographic signatures of lattice distortions within Kilauean olivines. a) EBSD
 maps colored using the IPF key as in Fig. 2. b) WBV directions overlain on band contrast
 maps ([100]=green, [010]=blue, [001]=red). c) Misorientation axes across subgrain
 boundaries located within the red box indicated in column a). Subgrain boundaries with [100]
 WBVDs (green) show misorientation axes between [001] and [010]. Subgrain boundaries
 with [001] WBVDs (red) show misorientation axes about [100] or [010].

**We thank the reviewer for their support of our method.
 We have added into the figure caption that the number bar on c represents (Lines 604-605):
 The color scale has units of ‘multiples of uniform distribution’.**

Figure 5

Fig. 5 – Chemical differences between deformed and undeformed olivine grains. a-b) Olivine
forsterite contents versus matrix glass Mg# ($Fe^{3+}/Fe_T=0.15$; ~QFM)³⁸. Olivine-liquid

equilibrium lines are shown for $K_d=0.270-0.352^{69,70}$. Both deformed and undeformed grains
have higher forsterite contents than the equilibrium olivine composition of their matrix
glasses. However, deformed grains show a narrower range of forsterite contents, which lie
further from the equilibrium field than undeformed grains. c-f) Cumulative distribution
functions comparing undeformed (solid lines) and deformed (dashed lines) olivines for 4
eruptions. Deformed grains are more primitive at a statistically significant level ($p<0.01$) using
the two sample Kolmogorov-Smirnov test. The test statistic D is also shown.

As the reviewer correctly points out, the range of Fo contents in deformed grains is
relatively large, but still far smaller than undeformed grains (e.g. compare Mauna Iki).

We have made the symbols larger, and adjusted the y axis as suggested.

We have added more detail regarding the intracaldera eruption sites into figure 1, 5
and the main text (Lines 125-126):

“Intracaldera summit eruptions (August, 1971, September 1971, July 1974, September
1974) contain no grains >1 mm, but all distorted grains are >840 μm ”.

We have combined these eruptions in terms of their symbol, as there are very few
deformed grains in eruption (we have only found one or two).

As discussed above, we still feel the sampling size per eruption is sufficient to see
that generally, deformed grains and undeformed grains have different chemistry. We
performed many analyses (82 crystals) on Episode 12 of Mauna Ulu while developing
the EBSD method, hence this eruption is extremely well categorized (and shows
statistically significant p values). While the N for the 1971 SWRZ eruption is low (due
to the relatively paucity of olivine in this sample), we still feel it is meaningful that
there is no overlap whatsoever between deformed and undeformed grains.

Additionally, the KS test takes the sample size into account, so the small sample size
is reflected in the relative high p value despite the clear separation of the populations.

We have added “between deformed and undeformed Kilauean olivines grains” into
the caption as suggested.

We have used the extreme values of olivine-melt K_d s due to the uncertainty regarding
what the correct value is at Kilauea (e.g. Helz et al., 2017 suggested that the average
K_d was 0.280, while Matzen et al., 2011 favoured 0.34).

We have clarified that the test statistic D quantifies the distance between two
distributions (Lines 623-624).

“The Kolmogorov-Smirnov test statistic D (representing the distance between the two
distributions) is also shown.”

513 Figure 6

514

32 grains were analysed by EBSD from the ~1700 AD flow, four of which overlap with
Hualalai in terms of Ca contents. Again, while this is a relatively small number of
grains, this is the only sample containing grains with this low Ca contents (compare
to blue lines).

We have removed the log vertical scale, and added an approximate horizontal scale following the reviewer suggestion. We have also lightened the color scheme. We thank the reviewer for pointing out the missing a) label on the figure.

We have made the suggested changes to the text in the figure caption (Lines 678-679).

“Schematic cross section along the magmatic plumbing system of Kilauea diagram showing the formation of deformed olivines”

mush piles. Primary members can experience forces ~6x the mean force, secondary
members experience less than the mean force, while spectator particles experience no force.

Figure 10

We have changed the size of the font.

We feel the need to emphasize that this is a simplistic model, informed by the
available constraints on magma reservoir size and primary magma composition
available in the literature. We appreciate that the size of Kilauea's magma chamber is
still a matter of debate (as the volume and geometry is uncertain). We use INSAR
estimates in this study as these are the only constraints to our knowledge that
estimate the areal footprint of the chamber (which is more important than the volume
for the thickness of mush piles). Regarding the composition of the magma entering
the upper crust, we appreciate that this value is debated, as magmas with 16.5 wt%
MgO have not been sampled in subaerial eruptions. However, magmas with up to
14.72 wt% MgO have been sampled on the subaerial portion of Kilauea's ERZ. Given
that this study argues against deep rift zone bypass from the summit reservoir, it is
plausible that these compositions are present in the SC reservoir (and due to
mixing/density contrasts, rarely erupt in subaerial lavas). Furthermore, shards of the
Pahala ash contain 13-14.5 wt% MgO, suggesting that more primitive lavas are
present at the base of the summit reservoir (Helz et al. 2015). We specifically choose
16.5 wt% for consistency with the volume balance used in Clague and Denlinger
(1994; 14% olivine fractionation). Even if this estimate is slightly too high (by a few
778 wt%), the overall difference to the model results is overwhelmed by uncertainty in
reservoir radius, and the height of mush piles indicated by subgrain piezometry.

Reviewer 3 – David Wallis

Review of NCOMMS-19-09434 by Wieser et al.

General comments

The manuscript addresses the origin of distorted olivine grains erupted from Kīlauea, for
which a variety of disparate models have previously been proposed. This topic is important
in deciphering the magmatic and petrological processes operating in the deep portions of the
magmatic system, including the states, locations, movements, and evolution of partially
solidified magma bodies. Distorted olivines are common features of mafic magmatic systems
worldwide and therefore Kīlauea provides an important example of phenomena and
processes that may be common. The authors take a new approach to characterising the
olivines by employing electron backscatter diffraction (EBSD) to map their internal
substructure, generating rich, quantitative datasets that go beyond those obtained previously
by other techniques. The results point to a new model for the generation of distorted olivines
involving deformation in crystal mush piles. The new model reconciles previously disparate
petrological, geochemical, and geophysical observations and therefore the results will be of
interest to workers in all these areas. As such, the general topic is suitable for Nature
Communications. The EBD work is extensive and contains novel aspects. One example is
the automated incorporation of boundary traces into the classification of subgrain
boundaries. The interpretation of the results is deep and rigorous. I have highlighted below
several minor scientific points that would benefit from further clarification, but the overall
scientific quality of the work is excellent. Therefore, I expect that the manuscript will be
suitable for publication once the authors have had an opportunity to make minor revisions to
address these points. I have also included several minor, less scientific, suggestions for the
text, which are intended only to be helpful.

Sincerely, David Wallis

**We thank the reviewer for their helpful technical comments, and for their support of**
**the novel aspects of our EBSD methodology, and the manuscript as a whole.**

**Specific scientific comments**

L40: Should this passage be “where olivine is the only silicate phase in magmas containing
>6.8 wt% MgO”?

**We have amended this sentence to the suggested form for clarity.**

L76–79: You could be even more specific about the benefits of EBSD analysis. In particular,
you can analyse large numbers of subgrain boundaries to improve statistical significance.
This benefit is particularly potent in the determination of misorientation axes, which are
imprecise when the misorientation angle is small (Prior, 1999; Wilkinson 2001) and therefore
require large numbers of measurements.

**We have rephrased this paragraph to provide more detail into the quantitative nature**
**of EBSD, and have added a sentence addressing the point above (lines 521-526)**

*“Crucially, EBSD diffraction methods allow assessment of the misorientation and WBVDs for*
*lots of neighbouring pixels along a given low angle boundary, and large numbers of*
*individual low angle boundaries in relatively short acquisition times (10-15 mins per grain),*
*and processing times (~2 minute per grain). The large datasets generated using these*
*methods are crucial for the precise characterisation of microstructures in volcanic olivines*
*which show only subtle distortions.”*

L100: Replace “during textural re-equilibrium” with “during textural re-equilibration”.

**Amended**

L106: I'm not sure I understand the argument about truncation, which may be because I'm
not sure that 'truncation' is the right word. The current text sounds as though some of the P-
rich domain has been removed by the subgrain boundary, whereas it looks as though the
subgrain boundary just happens to coincide with the divide between a P-rich core and P-
poor rim. Perhaps some elaboration or rephrasing is necessary.

**We thank the reviewer for highlighting that this sentence needed clarifying. We have**
**amended it to read (Lines 140-145):**

*"One P-rich zone terminates against a subgrain boundary (Fig. 2d), suggesting that P*
*enrichments and subgrains did not form during the common process of dendritic growth.*
*Instead, the rapid growth episode generating the P zoning may have preceded the distortion*
*of the crystal lattice, perhaps separated by a dissolution episode (c.f., ref. ⁵). Thus, while*
*some olivines at Kīlauea may have experienced an early dendritic growth phase, this*
*process appears unrelated to the production of lattice distortions"*

Figures 3a: Why the difference in the colour key between Figure 3a and those of figures 2
and 4a?

**In figure 2 and 4a, the IPF color scheme is adjusted to emphasize the small**
**misorientations in distorted olivines (pixels misorientated by > 3 degrees are shown**
**as black, to allow the other colors to emphasize the low angle boundaries in volcanic**
**olivines). As dendritic branches show much larger misorientations (a crucial way in**
**which they differ from low angle boundaries), the color scale was stretched out to 20**
**degrees, so that the misorientations within dendrites could be visualized. We have**
**clarified this difference in the figure caption for Fig. 3a (Lines 586-588):**

*"To emphasize the larger range of misorientation in dendrite branches compared with the*
*distorted olivines in Fig. 2, the color scale was adjusted so that pixels with deviation angles*
*>20° are colored black"*

Figures 3c and 4c: State that the colour scales have units of 'multiples of uniform
distribution'.

**We have stated in the captions for Figure 3c and 4c that:**

*"The color scale has units of 'multiples of uniform distribution'."*

L139: It would be better to cite original deformation experiments than Boioli's modelling
paper. How about Bai and Kohlstedt (1992, Tectonophysics)?

**We thank the reviewer for this suggestion, we have replaced the Boioli reference with**
**Bai and Kohlstedt (1992)**

L193 onwards: The use of 'fabric' and 'CPO' could be clarified. You seem to be using
"fabric" for the standard/idealised/endmember CPOs identified by Karato, Jung, et al. and
"CPO" for other measured CPOs (e.g., "relationships between fabric types and
crystallographic preferred orientations (CPOs)"). However, this is somewhat of a false
dichotomy and further muddies the water regarding the (already ambiguous) term 'fabric'. I
would simply use CPO throughout and scrap fabric, which would be particularly beneficial in
distinguishing when you are talking about CPO versus other microstructural elements
indicative of particular slip systems, such as subgrain boundaries (which some workers
would also include as 'fabric elements', but in a different sense to the use of 'A-type fabric'
(meaning CPO) by Jung et al.).

**We thank the reviewer for pointing out this ambiguous use of terminology. We now**
**use “Microstructure”, and have taken more care separating CPO terminology from**
**slip system terminology.**

198: Lars’s paper on natural peridotites is not the best reference for a statement about
deformation experiments. You already cite another of his papers, which does focus on
Paterson experiments, so you could cite that here too instead.

**We have replaced Hansen and Warren (2015) with Hansen et al. (2014)**

L206: Lloyd et al. (1997, Tectonophysics) and de Kloe et al. (2002, Microscopy and
Microanalysis) would be more appropriate references for determining slip systems from
subgrain boundaries than Dave Prior’s EBSD review paper.

**Again, we thank the reviewer for improving our referencing, we have added the Lloyd**
**et al 1997 reference. We feel keeping the review paper is useful for readers who are**
**not EBSD experts.**

L209: “We propose that the proportions of different types of dislocation preserved in low
angle boundaries reflect the relative activities of different slip systems.” This isn’t really your
proposal. It’s an established idea that has been around for decades (e.g., references above),
which is better for the strength of your argument anyway, so this sentence should be
rephrased.

**We have significantly rephrased this section following comments from Reviewer 1**
**that the manuscript was not accessible to the non-microstructural community. We**
**address these points in the following lines (87-97, 108-113):**

*“If distorted olivines record plastic deformation, rather than crystal growth processes, their*
*microstructures can be evaluated within the conceptual framework developed through the*
*extensive study of naturally- and experimentally-deformed mantle peridotites. These studies*
*have identified five olivine fabric types (A to E-types) with different crystallographically*
*preferred orientations (CPOs), and determined the role that pressure, temperature,*
*differential stress and hydration exert on their formation^{29–31}. However, unlike thin sections of*
*mantle peridotites, the orientations of erupted olivines in tephra and spatter deposits have*
*been randomised upon eruption and again during sample preparation, so CPOs cannot be*
*used to infer deformation conditions in volcanic olivines. However, CPOs are macroscopic*
*features resulting from the activities of different dislocation types (which are classified in*
*terms of slip systems)^{30,32–35}.”*
...

*EBSD maps of individual olivines provides sufficient crystallographic information to quantify*
*the Burgers vector and slip plane of low angle boundaries (see methods). As many*
*deformation experiments were conducted at conditions similar to those found within igneous*
*plumbing systems (e.g., 250–300 MPa, 1200 °C)³⁷, assessment of the activities of different slip*
*systems places constraints on the conditions of deformation (using established links between*
*slip systems, CPOs and fabric types, and between fabric types and deformation*
*conditions)^{30,32–35}.”*

L214–215: Insert “edge dislocations”

**Inserted**

L215: Replace with “twist walls consisting of [100] and [001] screw dislocations”

**Inserted**

Figure 8a: I don't think it makes sense to refer to subgrain boundaries as "A-type", etc.

Those terms were invented for CPO and using them here muddies the water again. I would

wait until Figure 8c, in which you relate the slip systems to CPO types, instead.

**We have removed the A-E type labels on figure 8a.**

L260: I don't think you need to invoke the strain compatibility problem here. At relevant

temperatures, olivine is essentially viscous, rather than plastic (meaning that the yield

stress/CRSS is negligible). If you stress the crystal in an arbitrary orientation some strain

rate will be produced on each slip system in proportion to their resolved shear stresses and

viscosities.

**We thank the reviewer for this comment, we have removed this sentence.**

L281: Change to "distance between subgrain boundaries".

**Amended**

L297: Round the stress estimates.

**Have rounded to ~4-11 MPa**

L326: Replace "process" with 'phenomenon'.

**Amended**

L328: Replace "parameter space" (which can't generate mush piles) with 'time scales' or

similar.

**We have changed this sentence to read (Lines 386-389):**

*"To further assess the hypothesis that deformed grains form in mush piles accumulating at*

*the base of the SC reservoir, we explore the timescales required to generate mush piles of*

*the thicknesses indicated by subgrain piezometry (~180-720 m) using a simple box model*

*informed by available geophysical and geochemical constraints at Kilauea."*

L337: Replace "InSAR displacements ... can be modelled as penny shaped cracks" with

'can be modelled based on a magma chamber represented by a penny-shaped crack'.

**Amended**

L370: Replace "is thought to" with a statement of the evidence.

**We thank the reviewer for suggesting this improvement, we have now added that this**

**statement is based on the decoupling of N-S and E-W tilt data (Lines 368-370):**

*This supports the hypothesis that the Mauna Iki eruption tapped magma from multiple*

*reservoirs based on the decoupling of E-W and N-S tilt components²¹.*

Figures 2, 3a, and 4a: State that the colour is based on the misorientation axis of each point

relative to the mean orientation (if this is correct).

**We have clarified the color scheme in Fig. 2 to reflect these suggestions (Lines 569-**
**573):**

*“A color reference direction is chosen such that the mean orientation of the central grain is*
*coloured white. The coloring denotes the misorientation axis (and angle) of each pixel*
*relative to the mean orientation. For example, pixels misorientated about [010] by 3 degrees*
*from the mean orientation are a dark blue (see the color scale, top right)”.*

Text suggestions L10 onwards: comma after “e.g.”

**We have replaced e.g. with e.g., at all relevant places in the text.**

L33 onwards: Indicate a range of numbers with an en dash, rather than a hyphen.

**We have replaced all relevant hyphens with en dashes.**

L45: Update to “Materials with high seismic velocities”.

**Updated**

L47 onwards: Up to now you have used an Oxford comma to separate the final items in
lists, but you have not done so after “events”. I recommend continuing the Oxford comma as
it has no downsides and sometimes adds clarity.

**We have added Oxford commas in the relevant places**

L51: “south flank” should be hyphenated as a compound adjective. However, it would be
better to avoid making a new jargon term, “south-flank instability”, and instead say “instability
of the south flank”.

**We agree with the reviewer that making more Kilauea jargon is not needed, we have**
**changed this to “instability of the south flank”.**

L63 and other instances: Avoid using “This” as an indefinite antecedent (not saying what
‘this’ is). Update to “This suite of eruptions...” or similar. There are many instances of similar
indefinite antecedents throughout the manuscript.

**We have amended this sentence, and others like it.**

L73: “oxidation-decoration” only needs to be hyphenated when used as a compound
adjective (e.g., the oxidation-decoration technique), whereas here it does not.

**We thank the reviewer for highlighting this. We have removed the hyphen.**

L75 onwards: There’s no need to capitalise words to define an acronym.

**We have removed the capitals in ‘electron backscatter diffraction’.**

L112: “were examined”... Up to now you have written in active voice but here you switch to
passive voice. It’s better to remain in active voice as it’s more explicit (e.g., in this case who
did the examining).

**We have changed this sentence to read (Lines 148-150):**

*“We examine low angle subgrain boundaries in 137 distorted olivine crystals from Kīlauea,*
*and compare these to the crystallographic signature of dendritic branching in 61 olivine*
*dendrites from West Greenland (Larsen and Pedersen, 2000).”*

L124: Here, “low angle” is a compound adjective and therefore should be hyphenated.

**Amended**

L128: Replace “which” with ‘that’.

**Amended**

L167: “requires”

**Amended**

L198: comma after “that”

**Amended**

L222: “direction of the” is redundant.

**Amended**

L224: ‘within individual EBSD maps’

**Amended**

L236: Comma after “i.e.”

**Amended**

L275: Hyphenate the compound adjectives “subgrain-size and dislocation-density
piezometry”.

**Amended**

L277 onwards: Scalar variables should be italic.

**Amended**

L358: Replace “deformation” with ‘deformed’.

**Amended**

L361: Replace “thought to” with ‘for which Pb isotopes indicate that they’

**We have updated this sentence to read (Lines 360-362):**

*“We find the largest proportion of deformed olivines in eruptions derived from the deeper*
*South Caldera reservoir (based on Pb isotopes; Pietruszka et al., 2018, 2015).”*

L367: Delete “glasses”.
**Amended**

L406: No need to capitalise “Electron Backscatter Diffraction”

**Amended**

L479: Replace “misorientated” with ‘misoriented’.

**Amended at all relevant points**

**END OF REVIEW**

**Reviewer 4 –**

I reviewed the manuscript “Distorted Olivine Crystals: An unexploited record of magma
storage at Kīlauea Volcano, Hawai‘i” by Wieser et al. despite not being a particularly strong
expert in mineral fabrics and EBSD. Therefore, I would like to comment mostly in general
terms and recommend that a better expert reviews some of the technical aspects of this
manuscript in addition to me.

Overall, I was intrigued by this study and it may provide some new ways at looking at crystal
populations. It may complement geochemical tools and also help understand the emergence
of various geochemical signatures and trends.

**We thank the reviewer for their support of the novel methods used in our study.**

In some parts I find the use or importance a little overstated. E.g. these types of olivines are
multiple times connected to volcanoes in hotspot settings and arcs. However, the only
reference to the arc setting (Chile) is an abstract and therefore, it has yet to be seen whether
this is a relevant phenomenon for arc volcanoes. The references referring to field studies are
with a couple exceptions almost exclusively on Hawaii/Kilauea (to some part that is
understandable given the work being on Hawaii), but in the selection of references the
authors fail to demonstrate that this has broader implications beyond Iceland, Reunion and
Kilauea. The authors generalize already in the first paragraph that this “...accounts for the
presence of deformed grains in many basaltic volcanoes”. I am not convinced that they have
shown that broad importance in this study.

**We thank the reviewer for their comments. We have refocused the manuscript to**
**make it clearer that a) deformed olivines are present in a wide variety of basaltic**
**systems b) the methods presented here have broad applicability in a number of**
**volcanic systems (even including those which exhibit different crystal phases; Lines**
**215-220):**

*“A crucial observation providing insight into the formation of deformed olivines is the fact that*
*these features have been noted in a variety of basaltic systems. For example, Vinet et al.²⁴*
*report deformed olivines in all five of the volcanic centres which they examined in the*
*Andean Southern Volcanic Zone. Deformed olivines have also been noted in the Vaigat*
*formation of West Greenland²², the Baffin Bay volcanic province⁴⁷, Piton de la Fournaise*
*volcano⁵, and the Loch Uisg area, Mull⁴⁸. Unlike Kīlauea, these localities do not contain deep*
*rift zone dunites. This implies that olivines are deformed by a process which is near*
*ubiquitous in a wide range of volcanic plumbing systems”*

**Lines 448-458:**

*“Our study demonstrates that microstructural investigations of erupted olivine crystals by*
*EBSD generates rich datasets which provide quantitative insights into crystal storage within*
*mush piles. Under the increasingly prevalent view that crustal magmatic systems are mush-*
*dominated⁷⁴, constraining the geometry and dynamics of crystal storage regions is crucial to*
*further our understanding of magmatic plumbing systems. The presence of deformed*
*olivines in many different volcanic settings highlights the global applicability of the methods*
*developed in this study. Furthermore, assessments of deformation conditions using EBSD*
*need not be restricted to olivine-bearing samples. Microstructural fabrics types in natural and*
*experimental samples have been established for a wide variety of igneous phases (e.g.,*
*diopside⁷⁵, plagioclase⁷⁶, hornblende⁷⁷), extending the applicability of our study to more*
*evolved igneous systems. “*

**It is also worth noting that deformed grains likely exist in many more systems than**
**we summarize in the text. As stated by Vinet et al. 2015, “Very few studies have**
**examined olivine microstructures and intracrystalline deformation mechanisms within**
**a volcanic context, and none have examined subduction related volcanism”. However,**
**they find deformed crystals at all 5 centres that they examine within the SVZ,**
**suggesting that olivine deformation is also a ubiquitous process in primitive arc**
**basalts, as well as the OIBS which have been subject to more intensive petrological**
**study. The lack of reports of deformed grains at other subduction zone volcanoes**
**may reflect the fact that no one has looked for them, the focus of many subduction**
**geochemists on whole-rock chemistry (Pers Comms, S Turner, 2019) or the fact that**
**(until our paper), there was no explanation for how these features form in basaltic**
**volcanoes without deep rift zones. Authors have perhaps not mentioned them in their**
**papers because they could not comment on the physical processes leading to their**
**formation.**

**Regarding the occurrence of deformed olivines in hotspot volcanoes, we believe it is**
**no coincidence that these features have only been reported in Iceland, Reunion and**
**Hawaii, which represent the OIBS which have been most intensely studied from a**
**petrological standpoint. For each of these hotspots, deformed olivines have been**
**found in many suites of lavas from different volcanoes. This suggests that further**
**investigations at other hotspots would also find these features. For example, olivine**
**deformational features have been noted at Isla Santiago in the Central Galapagos by**
**M Gleeson (see figure below), but these features were not mentioned in the final**
**publication due to stringent word limits for the journal Geology (M Gleeson, personal**
**communication).**

This clearly demonstrates that the absence of evidence for deformed lavas in other basaltic volcanoes is not evidence for their absence. We hope that the EBSD method developed in our manuscript will allow future studies to assess the presence/absence of intracrystalline distortion in a much more rigorous way than has been possible previously, which will hopefully lead to more papers reporting these features when present.

Please motivate more clearly why you want to compare the Kilauea results with West Greenland Picrites (l.67-68); this sentence lacks any motivation and the reader has to infer why you want to do this.

We have clarified that we examine west Greenland Picrites to assess the dendritic growth hypothesis proposed by Welsch et al. 2013 (Lines 68-71):

“Finally, to evaluate the suggestion that lattice distortion results from the textural maturation of olivine dendrites, we assess the crystallographic signatures of branching dendritic growth in olivine dendrites from West Greenland picrites^{22,23}, and compare these to microstructures in deformed Kilauean olivines.”

I am missing a little more information on the size of the studied populations (Paragraph starting l.82). While the figures may reveal some of that info, I think it is important to say how many crystals were interrogated for each eruption and what the abundance of olivine is in these eruptions. Any reader would like to get a sense on whether we are talking 10s of grains in something that is crystal-rich? Or 100s of grains in a magma that is crystal-poor.

We thank this reviewer (and reviewer 2) for pointing out that we have not given sufficient information on the number of deformed grains examined in each eruption. We have now added the information on the number of deformed grains on each bar chart on figure 2.

Some of that information is presented, but not systematically. In this regard, size information is given (larger or smaller than 1 mm). How small are the smallest crystals that were investigated? Could this be presented in the context of crystal size distributions and are different populations inferred from those?

We have been vague about the size information because these crystals were recovered from jaw-crushed tephra, so the measured size is always a minimum of the true size. Also, during the picking and mounting process, it is extremely easy to bias the chosen crystals towards larger grain sizes. Thus, it is not possible to generate crystal size distributions from our data. We split grains into >1mm and <1mm based on the sizes of the sieves used during sample preparation. This proves that, generally, larger crystals are more likely to be deformed, however, we do not feel we can quantify this further on the current sample set.

l. 206f: The authors point to previous work in this sentence, but don't have a citation. Is citation 44 (cited just before that) implied? Consider reorganization.

We have significantly rephrased this section following other reviewer comments, but have clarified the citation.

The mush pile model (l.309) is making a lot of assumptions. I wondered whether these dislocations are not only a product of the overall stress, but whether there is a time-dependence on how long these grains are exposed to those loading stresses. Also, if the stress disappears (e.g. because the grain is moving from a stress chain setting to an unloaded configuration) do these dislocations also disappear through local equilibration in the lattice?

The time dependent evolution of dislocations within olivine crystals is poorly resolved. While it is logical that dislocations would heal as stresses reduce, the geological record is full of rocks under zero stress that contain dislocations. In all likelihood, the average dislocation density we measure may be overprinted by a late stage stress-free anneal. However, the time dependent behaviour of subgrains (which we predominantly use to obtain our stress estimates) is highlighted by Poirier, 1985 (Poirier, J. P. 1985. Creep of Crystals. Cambridge University Press, Cambridge):

“ It is also generally accepted that the subgrain size varies rapidly during a stress increase, but is stable against a stress decrease, hence it would be representative of the maximum stress experienced by the mineral.”

Additionally, Helz (1987) compare deformation textures in eruption pumices from Kilauea Iki with lavas which had ponded, and remained at high temperatures for >22 years. They find that these two olivine populations show no differences, supporting the statement of Poirier that extinction discontinuities have not been removed by annealing. Due to the highly speculative nature of this topic, we have not added further detail into the manuscript.

After all unlike in situations of truly high crystal volume fraction situations (e.g. dunite) the time that an individual crystal spends loaded may be intermittent with healing episodes in between.

We agree that the migration of force chains will lead to time-varying stresses. However, we have no real constraints on the rapidity of force chain migrations, or the rate of dislocation healing in natural samples (other than that discussed above).

Another aspect in this regard, what is the actual residence time of a distorted olivine in the mush pile? Especially within the lower layers where the full >500 m mush pile is resting on the crystals? I understand that these questions are difficult to answer, but I would greatly appreciate a discussion about these temporal and spatial complications. A short discussion in I.385 is provided, but the authors basically punts on this question.

Minimum estimates of storage times within mush piles are provided by melt inclusion records (addressed in a second manuscript currently in revision for Nature Communications). Figure 2 from this paper is shown below. Melt compositions at Kilauea volcano show prominent cycles over times. Eruptions containing abundant deformed olivines (and highly forsteritic olivines, panel c) have melt inclusion populations spanning very broad ranges in Nb/Y, larger than those observed in melt compositions since 1800, and more comparable to the range of whole rock compositions over 350kyrs. This, along with the similarity of Nb/Y ratios in many different eruptions provides further evidence that crystals are scavenged from mush piles accumulating over timescales longer than a few centuries. We have referenced this paper (Line 399-401):

“This is consistent with the diverse range of trace element ratios within melt inclusions in any given eruption, which require mush pile residence times of >170 years⁷⁰.”

Regarding the timescales upon which olivines begin to migrate downrift, it is almost impossible to place constraints on this process. The inherent cyclicality of geochemical variations means that it is not currently possible to estimate maximum storage timescales. While the range of Nb/Y present in melt inclusion populations could have accumulated over ~170 years, it could also have accumulated of 350 kyrs (as melt inclusions in some eruptions contain as much diversity as a 350kyr record of erupted lavas). Until a geochemical tracer is identified which shows unidirectional changes,

and can be analysed in-situ in melt inclusions, we are afraid that placing further
constraints on storage timescales is not possible.

REVIEWERS' COMMENTS:

Reviewer #1 (Remarks to the Author):

Regarding manuscript NCOMMS-19-09434A-Z re-review:

The authors have addressed all of my concerns more than adequately! I'm happy with the new version of the manuscript.

And I love the new supplementary section comparing the optical images to the EBSD maps! It makes a nice comparison, and a convincing argument for additional resolution supplied by EBSD maps.

The crystallographic axes have been indicated nicely, and the confusion over crystallographic axes (a,b,c) and specimen coordination system (X,Y,Z) has been resolved.

Thank you for switching to a Right Handed System for your IPFs!

The only changes I would make at this point are as follows:

In Figure 3, the IPF scale on the left hand side is murky with grey apices - this should be replaced if possible. The one on the right hand side is fine.

Also, I noticed that a few times in the text there is still no space between a value and its units.

E.g. line 341: 720m

line 491: ~40um

line 559: >1mm and <1mm

Be sure to check for more of these, as I have not provided a complete list.

Nicely done!

I recommend publication!

Roberta Flemming

Reviewer #2 (Remarks to the Author):

I am completely satisfied with this revised version of the Wieser et al. manuscript entitled "Distorted Olivine Crystals: a record of magma storage at Kīlauea Volcano, Hawai'i".

You charged me with: "In particular, we were very concerned about the overlapping novelty with this study and that of Bradshaw et al., however, the authors have now refocused the manuscript to highlight the novelty in the new application of the EBSD method to magmatic systems. We hope you will be willing to look at the point by point response letter and the revised manuscript, and tell us whether you feel that the points raised in the previous round of review have been satisfactorily addressed. In particular, we would be grateful to hear your comments on the advance of this new application of the EBSD method to igneous petrology."

1. I was involved in the Bradshaw et al. paper as indicated by their citing me for help in their acknowledgements. I feel the Wieser et al. paper is a leap forward in the application of EBSD techniques to understand the history of olivine in magmatic systems.

2. Point by point response letter assessment. The authors have done a thorough job in responding to all of my comments and questions. I am totally satisfied with their responses and with the changes they made to the paper, especially improving the figures and table.

Other comments.

The interpretation of this study will stimulate discussion in the volcanological and petrologic communities. I hope other methods (e.g., CO₂ in melt inclusions) will be used to test their hypothesis of deformation in the mush pile under the summit caldera. However, the samples used for this study were all taken near the summit of Kīlauea; thus they may be correct in their

interpretation for these samples. However, the paper makes inferences about rift zone processes which are more tenuous. The studies by Garcia et al. (1989) and Clague et al. (1995) show that picrities (>15 vol.% olivine) are exceedingly common along the submarine rift zones of Hawaiian volcanoes including Kilauea. I doubt whether all of the olivine in rift zone lavas originated in the summit mush pile given the volumetric constraints imposed by the reservoir system (e.g., Pietruszka, et al., 2015. Two small magma bodies beneath the summit of Kilauea Volcano unveiled by isotopically distinct melt deliveries from the mantle. *Earth and Planetary Science Letters*, 413, 90–100, doi.org/10.1016/j.epsl.2014.12.040.). Nevertheless, this study offers a testable hypothesis which is what is needed to move forward on these fundamental issues.

I am surprised and delighted to see they have used Hawaiian punctuation properly and consistently. This is not common for those who live outside of Hawai'i.

Reference 14. Garcia, M. O. A Petrologic Perspective of Kilauea Volcano's Summit Magma 759 Reservoir. *Journal of Petrology* 44, 2313–2339 (2003).

Should be Garcia, M. O., Pietruszka, A.J. & Rhodes, J.M.

If the reviews get posted with this paper, please change anonymous to M. O. Garcia

Reviewer #3 (Remarks to the Author):

The authors have revised the manuscript substantially to deal thoroughly with the points raised by myself and other reviewers during the first round of reviewing.

I was asked to comment in particular on the significance of the advance of this new application of the EBSD method to igneous petrology, and specifically, whether I think the method has potential to be applied to magmatic systems beyond Hawaii and intra-plate volcanoes. In this respect, I see that the manuscript has three main strengths:

1) As I stated in my original review, this manuscript reports novel data processing that combines weighted Burgers vector directions and boundary traces in a new way that is beneficial to the interpretation of subgrain-boundary types. This development is useful in applications to igneous petrology but also much more widely in the study of deformed rocks (consisting of multiple minerals) more generally.

2) My impression is that, although EBSD has been employed in several recent papers on igneous petrology, the full power of the technique has generally not been exerted in this field. As this manuscript is the product of an unusually extensive and detailed body of work, I anticipate that the paper will stimulate interest in exploiting the full potential of EBSD in a variety of igneous-petrology problems.

3) Regarding the generality of the occurrence of deformed magmatic olivines and the potential to analyse them using the documented approach, the authors make a strong case in the revised manuscript that such microstructures are common in a variety of settings and, if anything, are likely to be more common than is currently appreciated. There is no technical/methodological reason why the approach presented in the manuscript cannot be applied to those microstructures.

In summary, the authors have done a good job at dealing with the reviewers' comments. The manuscript presents work that is both novel and thorough and has provocative implications for the processes operating in the deep portions of magmatic plumbing systems. The method has applicability beyond the immediate field of the manuscript and the results have implications for the interpretation of other geochemical and geophysical datasets pertaining to magmatic processes beneath volcanic edifices.

Sincerely,

David Wallis

Reviewer 1

The only changes I would make at this point are as follows:

In Figure 3, the IPF scale on the left hand side is murky with grey apices - this should be replaced if possible. The one on the right hand side is fine.

We have replaced the IPF key – We had originally tried to demonstrate how the IPF key is contracted to 20 degrees of scaling, but hopefully this is clear from the figure caption

“To emphasize the larger range of misorientation in dendrite branches compared with the distorted olivines in Fig. 2, the color scale was adjusted so that pixels with deviation angles >20° are colored black”

Also, I noticed that a few times in the text there is still no space between a value and its units.

E.g. line 341: 720m

line 491: ~40um

line 559: >1mm and <1mm

Be sure to check for more of these, as I have not provided a complete list.

We have added spaces at all relevant points in the text.

Nicely done!

I recommend publication!

Roberta Flemming

Reviewer 2

Reference 14. Garcia, M. O. A Petrologic Perspective of Kilauea Volcano’s Summit Magma 759 Reservoir. Journal of Petrology 44, 2313–2339 (2003).

Should be Garcia, M. O., Pietruszka, A.J. & Rhodes, J.M.

We have amended this referencing mistake.

Reviewer 3

No corrections required